# OFF-POLICY REINFORCEMENT LEARNING WITH DELAYED REWARDS

## ABSTRACT

We study deep reinforcement learning (RL) algorithms with delayed rewards. In many real-world tasks, instant rewards are often not readily accessible or even defined immediately after the agent performs actions. In this work, we first formally define the environment with delayed rewards and discuss the challenges raised due to the non-Markovian nature of such environments. Then, we introduce a general off-policy RL framework with a new $\mathcal{Q}$-function formulation that can handle the delayed rewards with theoretical convergence guarantees. For practical tasks with high dimensional state spaces, we further introduce the HC-decomposition rule of the $\mathcal{Q}$-function in our framework which naturally leads to an approximation scheme that helps boost the training efficiency and stability. We finally conduct extensive experiments to demonstrate the superior performance of our algorithms over the existing work and their variants.

## 1 INTRODUCTION

Deep reinforcement learning (RL) aims at maximizing the cumulative reward of a MDP. To apply RL algorithms, the reward has to be given at every state-action pair in general (*i.e.*, $r(s, a)$). With a good and high quality reward function, RL can achieve remarkable performance, e.g. AlphaGo Zero for Go (Silver et al., 2017), DQN(Mnih et al., 2015) for Atari, SAC (Haarnoja et al., 2018b) for robot control e.t.c. Recently, RL has been applied in many other real world settings beyond games and locomotion control. This includes industrial process control (Hein et al., 2017), traffic optimization (Gong et al., 2019; Lin et al., 2018), molecular design (Olivecrona et al., 2017) and resource allocation (Xu et al., 2018). However, in many of these real-world scenarios, Markovian and instant per-step rewards are hard or impossible to obtain or even clearly defined. In practice, it becomes more reasonable to use a delayed reward of several consecutive steps as feedback. For example, in traffic congestion reduction (Gong et al., 2019), the amount of decreased congestion for a single traffic light switch is hard to define in practice while it is more adequate to use the average routing time consumed for the vehicles as the feedback. The latter is a delayed reward which can only be obtained after the vehicles have passed the congestion (long after a single switch). In molecular design, only the molecule formed after the final operation can provide a meaningful evaluation for the whole process (Olivecrona et al., 2017). In locomotion control where the feedback is generated by the interaction with the large environment, it is usually the case that the frequency of the rewards generated from the environment's sensors is much lower than the frequency of the robot control, thus the rewards are given only every so often.

Despite the importance and prevalence of the delayed reward setting, very few previous research in RL has been focusing on problems with non-Markovian properties and delayed rewards. Besides, current RL algorithms lack theoretical guarantee under non-Markovian rewards and perform unsatisfactorily in practice (Gangwani et al., 2020). Thus, in this paper, our goal is to better understand the properties of RL with delayed rewards and introduce a new algorithm that can handle the delayed reward both theoretically and practically. A key to our approach lies in the definition of the past-invariant delayed reward MDPs. Based on this definition, theoretical properties and algorithmic implications are discussed to motivate the design of a new practical algorithm, which explicitly decompose the value function into components of both historical (H) and current (C) step information. We also propose a number of ways to approximate such HC decomposition, which can be readily incorporated into existing off-policy learning algorithms. Experiments demonstrate that such approximation can improve training efficiency and robustness when rewards are delayed.

## 2 PROBLEM FORMULATION

In order to characterize the delayed and non-Markovian reward signals, we introduce the Delayed Reward Markov Decision Process (DRMDP). In this work, we focus on DRMDP that satisfies the *Past-Invariant (PI)* condition, which is satisfied in many real-world settings with delayed rewards.

### 2.1 PAST-INVARIANT DELAYED REWARD MDPS

In DRMDP, the transition of the environment is still Markovian and the agent can observe and interact with the environment instantly. However, rewards may be non-Markovian and delayed and are observed only once every few steps. More specifically, the time steps are divided into consecutive *signal intervals* of random lengths, and the reward signal generated during a signal interval may depend on the state-action sequence and is observed only at the end of the interval. We formally define DRMDP as follows.

**Definition 1** (DRMDP). *A Delayed Reward Markov Decision Process $M = (S, A, p, q_n, r, \gamma)$ is described by the following parameters.*

1. *The state and action spaces are $S$ and $A$ respectively.*

2. *The Markov transition function is $p(s'|s, a)$ for each $(s, a) \in S \times A$; the initial state distribution is $p(s_0)$.*

3. *The signal interval length is distributed according to $q_n(\cdot)$, i.e., for the $i$-th signal interval, its length $n_i$ is independently drawn from $q_n(\cdot)$.*

4. *The reward function $r$ defines the expected reward generated for each signal interval; suppose $\tau_i = \tau_{t:t+n_i} = (s, a)_{t:t+n_i} = ((s_t, a_t), \ldots (s_{t+n-1}, a_{t+n_i-1}))$ is the state-action sequence during the $i$-th signal interval of length $n_i$, then the expected reward for this interval is $r(\tau_i)$.*

5. *The reward discount factor is $\gamma$.*

In this work, we focus on the infinite-horizon DRMDP. We use $(\tau, n) = (\tau_i, n_i)_{i \in \{1,2,3,\ldots\}}$ to denote a trajectory, where $\tau_i$ is the state-action sequence during the $i$-th signal interval and $n_i$ is the corresponding length. We also let $t_i$ be first time step of the $i$-th signal interval, *i.e.*, $t_i = \sum_{j=1}^{i-1} n_j$. Note that the reward $r(\tau_i)$ is revealed at time $t_i + n_i - 1 = t_{i+1} - 1$. We finally define the discounted cumulative reward of the trajectory $(\tau, n)$ by $R(\tau, n) := \sum_{i=1}^{\infty} \gamma^{t_{i+1}-1} r(\tau_i)$. The objective for DRMDP is to learn a policy $\pi$ that maximized the expected discounted cumulative reward $\mathcal{J}(\pi) := \mathbb{E}_{(\tau,n)\sim\pi}[R(\tau, n)]$.

**Augmented Policy Class.** In DRMDP, the Markov policy class $\{\pi = \pi(a_t|s_t)\}$ might not achieve satisfactory performance because of the non-Markovian nature of the reward function. Theoretically speaking, we need a more general policy class $\Pi_\tau = \{\pi = \pi(a_t|\tau_{t_i:t} \circ s_t)\}$ ($i$ is the index of the signal interval that $t$ belongs to). A formal statement is included in Appendix B. Unfortunately, policy optimization in such an exponentially large space $\Pi_\tau$ is hard in practice, especially for high-dimensional problems. Thus, as a trade-off, we consider the policy class $\Pi_s = \{\pi = \pi(a_t|s_t, t - t_i)\}$, which resembles the traditional Markov policy class, but is augmented with an extra parameter indicting the relative index of the current time step in the signal interval. Moreover, we focus on the PI-DRMDP problems, in which our algorithmic framework has theoretical guarantees.

**Definition 2** (PI-DRMDP). *A Past-Invariant Delayed Reward Markov Decision Process is a DR-MDP $M = (S, A, p, q_n, r, \gamma)$ whose reward function $r$ satisfies the following Past-Invariant (PI) condition: for any two trajectory segments $\tau_1$ and $\tau_2$ of the same length, and for any two equal-length trajectory segments $\tau_1'$ and $\tau_2'$ such that the concatenated trajectories $\tau_a \circ \tau_b'$ are feasible under the transition dynamics $p$ for all $a, b \in \{1, 2\}$, it holds that*

$$r(\tau_1 \circ \tau_1') > r(\tau_1 \circ \tau_2') \iff r(\tau_2 \circ \tau_1') > r(\tau_2 \circ \tau_2').$$

Roughly speaking, in PI-DRMDP, the relative credit for different actions at each $s_t$ only depends on the experience in the future and is invariant of the past. This property may relieve the agent from considering the past experience for decision making. A simple example of $r$ with PI condition is $r(\tau_{t:t+n}) = \sum_{i=t}^{t+n-1} \hat{r}(s_i, a_i)$, where $\hat{r}$ is a per-step reward function. This kind of tasks is studied in many previous work (Zheng et al., 2018; Oh et al., 2018; Klissarov & Precup, 2020). We refer this kind of reward functions as the sum-form.

In the worst case, there exists some bizarre reward design in PI-DRMDPs that still requires the optimal policy to take history information into consideration (see Appendix B). Despite the suboptimality issue, we will show that our algorithmic framework guarantees policy improvement in $\Pi_s$, which is sufficient to support the design of actor-critic methods. Consequently, we are able to put forward a practical RL algorithm that can achieve the SOTA performance in delayed-reward tasks. The overall contribution of our work is summarized in Table 1 and we leave the optimality of the DRMDP problems as future works.

Table 1: Contribution of our algorithm in different problem settings. Notice that MDP is a degraded case of DRMDP, which has signal interval length fixed as 1.

|          | Policy Improvement | Policy Optimality |
|----------|--------------------|--------------------|
| MDP      | Yes                | Yes                |
| PI-DRMDP | Yes                | No                 |
| DRMDP    | No                 | No                 |

**General Reward Function.** In Definition 1, we define the reward as a function of the state-action sequence of its signal interval. In general, we may allow reward functions with longer inputs which overlap with the previous signal intervals, e.g., maximal overlapping of $c$ steps, $r(\tau_{t_i-c:t_i+n_i})$. The theoretical analysis in Section 3.1 and the empirical method in Section 3.2 can be directly extended to this general reward function. We provide detailed discussions in Appendix B for the general definition while we only consider $c = 0$ in the main text for the simplicity of the exposition.

## 2.2 OFF-POLICY RL IN PI-DRMDP

Deep Reinforcement Learning has achieved great success in solving high-dimensional MDPs problems, among which the off-policy actor-critic algorithms SAC (Haarnoja et al., 2018a) and TD3 (Fujimoto et al., 2018) are the most widely used ones. However, since the rewards are delayed and non-Markovian in PI-DRMDP, directly applying these SOTA off-policy RL algorithms faces many challenges that degrade the learning performance. In this subsection, we briefly discuss the problems that arise in critic learning based on TD3 and similar problems also exist for SAC.

First, in PI-DRMDP, value evaluation with off-policy samples brings in off-policy bias. TD3 minimizes over $\phi$ w.r.t.

$$L_\phi = \mathbb{E}_D \left[ (R_t + \gamma \hat{Q}_\phi(s_{t+1}, a'_{t+1}) - Q_\phi(s_t, a_t))^2 \right], \tag{1}$$

where $(s_t, a_t, R_t, s_{t+1})$ is sampled from the replay buffer $D$, $R_t = r(\tau_i)$ if $t = t_{i+1} - 1$ and $i = i(t)$ is the index of the reward interval that $t$ belongs to, and $R_t = 0$ otherwise. $\hat{Q}_\phi$ represents the target value of the next state and $a'_{t+1}$ is sampled from the smoothed version of the policy $\pi$. Furthermore, in practical implementation, samples in the replay buffer $D$ are not sampled from a single behavior policy $\beta$ as in traditional off-policy RL (Sutton & Barto, 2018). Instead, the behavior policy changes as the policy gets updated. Thus, we assume that samples are collected from a sequence of behavior policies $\boldsymbol{\beta} = \{\beta_k\}_{k=1}^K$.

In the delayed reward setting, since the reward $r(\tau_i)$ depends on the trajectory of the whole signal interval (*i.e.*, $\tau_i$) instead of a single step, the function $Q_\phi(s_t, a_t)$ learned via Eq. (1) will also have to depend on the trajectory in the signal interval upto time $t$ (*i.e.*, $\tau_{t_i:t}$) rather than the single state-action pair at time $t$. Since the samples are collected under a sequence of behavior polices $\boldsymbol{\beta}$, different behavior policy employed at state $s_t$ may lead to different distribution over $\tau_{t_i:t}$ in Eq. (1). Consequently, $Q_\phi(s_t, a_t)$ will be affected by this discrepancy and may fail to assign the accurate expected reward for the current policy. Please refer to Appendix B for detailed discussions.

Second, in addition to the issue resulted from off-policy samples, $Q_\phi(s_t, a_t)$ learned in Eq. (1) with on-policy samples in $D$ may still be problematic. We formally state this problem via a simple sumform PI-DRMDP in Appendix B whose optimal policy is in $\Pi_s$. In this example, the fix point of Eq. (1) fails to assign the actual credit and thus misleads policy iteration even when the pre-update policy is already the optimal. We refer this problem as the fixed point bias.

Last but not least, the critic learning via Eq. (1) would suffer a relative large variance. Since $R_t$ varies between 0 and $r(\tau_i)$, minimization of TD-error ($L_\phi$) with a mini-batch of data has large

variance. As a result, the approximated critic will be noisy and relative unstable. This will further effect the policy gradients in Eq. (2).

$$\nabla_\theta J(\pi_\theta) = \mathbb{E}_D \left[ \nabla_{a_t} Q_\phi(s_t, a_t) \Big|_{\pi_\theta(s_t)} \nabla_\theta \pi_\theta(s_t) \right] \tag{2}$$

To sum up, directly applying SOTA off-policy algorithms in PI-DRMDP will suffer from multiple problems (off-policy bias, fixed point bias, large traning noise, etc). Indeed, these problems result in a severe decline in performance even in the simplest task when $n = 5$ (Gangwani et al., 2020).

## 3 METHOD

In this section, we first propose a novel definition of the $\mathcal{Q}$-function (in contrast to the original $Q$-function) and accordingly design a new off-policy RL algorithm for PI-DRMDP tasks. This method has better theoretical guarantees in both critic learning and policy update. We then introduce a HC-*decomposition framework* for the proposed $\mathcal{Q}$-function, which leads to easier optimization and better learning stability in practice.

### 3.1 THE NEW $\mathcal{Q}$-FUNCTION AND ITS THEORETICAL GUARANTEES

Since the non-Markov rewards make the original definition of $Q$-function ambiguous, we instead define the following new $\mathcal{Q}$-function for PI-DRMDP tasks.

$$\mathcal{Q}^\pi(\tau_{t_i:t+1}) := \mathbb{E}_{(\tau,n)\sim\pi} \left[ \sum_{j=i}^\infty \gamma^{t_{j+1}-t-1} r(\tau_j) \Big| \tau_{t_i:t+1} \right], \tag{3}$$

The new $\mathcal{Q}$-function is defined over the trajectory segments $\tau_{t_i:t+1}$, including all previous steps in $s_t$'s signal interval. Besides, the expectation is taken over the distribution of the trajectory $(\tau, n)$ that is due to the randomness of the policy, the randomness of signal interval length and the transition dynamics. Despite the seemingly complex definition of $\mathcal{Q}$, we provide a few of its nice properties that are useful for PI-DRMDP tasks as follows. The proofs are listed in Appendix B.

First, we consider the following objective function

$$\mathcal{L}_\phi := \mathbb{E}_D \left[ (R_t + \gamma \hat{\mathcal{Q}}_\phi(\tau_{t_j:t+2}) - \mathcal{Q}_\phi(\tau_{t_i:t+1}))^2 \right], \tag{4}$$

where $\tau_{t_j:t+2} = \tau_{t_i:t+1} \circ (s_{t+1}, a'_{t+1})$ (so that $j = i$ if $t$ is not the last step of $\tau_i$, and $\tau_{t_j:t+2} = (s_{t_{i+1}}, a'_{t_{i+1}})$ (so that $j = i+1$) otherwise. Similarly to Eq. (1), in Eq. (4), $(\tau_{t_i:t+1}, R_t, s_{t+1})$ is also sampled from $D$ and $a'_{t+1}$ is sampled from $\pi$ correspondently. We may view Eq. (4) as an extension of Eq. (1) for $\mathcal{Q}^\pi$. However, with these new definitions, we are able to prove the following fact [1].

**Fact 1.** *For any distribution $D$ with non-zero measure for any $\tau_{t_i:t+1}$, $\mathcal{Q}^\pi(\tau_{t_i:t+1})$ is the unique fixed point of the MSE problem in Eq. (4). More specifically, when fixing $\hat{\mathcal{Q}}_\phi$ as the corresponding $\mathcal{Q}^\pi$, the solution of the MSE problem is still $\mathcal{Q}^\pi$.*

Consequently , $\mathcal{Q}^\pi$ can be found precisely via the minimization for tabular expressions. This helps to solve the problems in critic learning for PI-DRMDP. We next introduce Fact 2 which states that the order of $\mathcal{Q}^\pi$ w.r.t. the actions at any state $s_t$ is invariant to the choice of $\tau_{t_i:t}$, thanks to the PI condition.

**Fact 2.** *For any $\tau_{t_i:t}, \tau'_{t_i:t}$ and $s_t, a_t, a'_t$ which both $\tau_{t_i:t} \circ s_t$ and $\tau'_{t_i:t} \circ s_t$ are feasible under the transition dynamics, $\forall \pi \in \Pi_s$, we have that*

$$\mathcal{Q}^\pi(\tau_{t_i:t} \circ (s_t, a_t)) > \mathcal{Q}^\pi(\tau_{t_i:t} \circ (s_t, a'_t)) \iff \mathcal{Q}^\pi(\tau'_{t_i:t} \circ (s_t, a_t)) > \mathcal{Q}^\pi(\tau'_{t_i:t} \circ (s_t, a'_t)).$$

Fact 2 ensures that the policy iteration on $\pi_k(a_t|s_t, t-t_i)$ with an arbitrary $\mathcal{Q}^\pi(\tau_{t_i:t} \circ (s_t, a_t))$ results in the same $\pi_{k+1} \in \Pi_s$. Consequently, we are able to prove the convergence theorem for the policy iteration with the $\mathcal{Q}$-function.

---

[1]The definition of $\mathcal{Q}^\pi$ and Fact 1 also holds in DRMDP and for $\pi \in \Pi_\tau$.

**Proposition 1. (Policy Improvement Theorem for PI-DRMDP)** *For any policy $\pi_k \in \Pi_s$, the policy iteration w.r.t. $\mathcal{Q}^{\pi_k}$ produces policy $\pi_{k+1} \in \Pi_s$ such that $\forall \tau_{t_i:t+1}$, it holds that*

$$\mathcal{Q}^{\pi_{k+1}}(\tau_{t_i:t+1}) \geq \mathcal{Q}^{\pi_k}(\tau_{t_i:t+1}),$$

*which implies that $\mathcal{J}(\pi_{k+1}) \geq \mathcal{J}(\pi_k)$.*

Thus, the off-policy value evaluation with Eq. (4) and the policy iteration in Proposition 1 guarantee that the objective function is properly optimized and the algorithm converges. Besides, for off-policy actor critic algorithms in the continuous environment, the policy gradient $\nabla_\theta \mathcal{J}(\pi_\theta)$ is changed to

$$\nabla_\theta \mathcal{J}(\pi_\theta) = \mathbb{E}_D \left[ \nabla_{a_t} \mathcal{Q}^\pi(\tau_{t_i:t} \circ (s_t, a_t)) \Big|_{\pi_\theta(s_t, t-t_i)} \nabla_\theta \pi_\theta(s_t, t-t_i) \right], \tag{5}$$

where $(\tau_{t_i:t}, s_t)$ is sampled from $D$. The full algorithm is summarized in Algorithm 1. As a straightforward implementation, we approximate $\mathcal{Q}^\pi(\tau_{t_i:t+1})$ with a GRU network (Cho et al., 2014) and test it on the continuous PI-DRMDP tasks. As shown in Figure 2(a), our prototype algorithm already outperforms the original counterparts (*i.e.*, SAC) on many tasks.

### 3.2 THE HC-DECOMPOSITION FRAMEWORK

One challenge raised by the $\mathcal{Q}$-function is that it takes a relatively long sequence of states and actions as input. Directly approximating the $\mathcal{Q}$-function via complex neural networks would suffer from lower learning speed (e.g., recurrent network may suffer from vanishing or exploding gradients (Goodfellow et al., 2016)) and computational inefficiency. Furthermore, the inaccurate estimation of $\mathcal{Q}^\pi(\tau_{t_i:t+1})$ will result in inaccurate gradients in Eq. (5) which degrades the learning performance.

To improve the practical performance of our algorithm for high dimensional tasks, we propose the HC-*decomposition framework* (abbreviation of History-Current) that decouples the approximation task for the *current step* from the *historical trajectory*. More specifically, we introduce $H_\phi$ and $C_\phi$ functions and require that

$$\mathcal{Q}_\phi(\tau_{t_i:t+1}) = H_\phi(\tau_{t_i:t}) + C_\phi(s_t, a_t), \tag{6}$$

Here, we use $C_\phi(s_t, a_t)$ to approximate the part of the contribution to $\mathcal{Q}_\phi$ made by the *current step*, and use $H_\phi(\tau_{t_i:t})$ to approximate the rest part that is due to the *historical trajectory* in the signal interval. The key motivation is that, thanks to the Markov transition dynamics, the current step $(s_t, a_t)$ has more influence than the past steps on the value of the future trajectories under the given policy $\pi$. Therefore, in Eq. (6), we use $C_\phi$ to highlight this part of the influence. In this way, we believe the architecture is easier to optimize, especially for $C_\phi$ whose input is a single state-action pair.

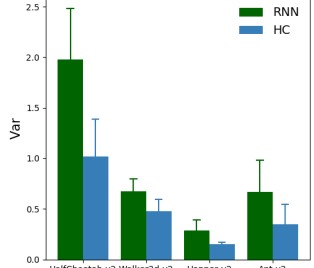

Moreover, we also find that the policy gradient will only depend on $C_\phi(s_t, a_t)$. Indeed, we calculate that the policy gradient $\nabla_\theta \mathcal{J}^{\text{HC}}(\pi_\theta)$ equals to

$$\mathbb{E}_D \left[ \nabla_{a_t} C_\phi(s_t, a_t) \Big|_{\pi_\theta(s_t, t-t_i)} \nabla_\theta \pi_\theta(s_t, t-t_i) \right], \tag{7}$$

Figure 1: Policy gradient variance averaged over the training process. All bars show the mean and one standard deviation of 4 seeds.

Comparing Eq. (7) with Eq. (5), we note that the gradient on the policy $\pi$ under HC-decomposition will not be effected by $\tau_{t_i:t}$. Thus, for a mini-batch update, our policy gradient has less variance and the training becomes more efficient. In Figure 1, we visualize and compare the scale of the gradient variance of our HC-decomposition framework and the straightforward recurrent network approximation of the $\mathcal{Q}$ function, w.r.t. the same policy and the same batch of samples. Clearly, the result supports our claim that HC-decomposition results in less gradient variance.

Finally, to learn the $H_\phi$ and $C_\phi$ functions, we minimize the following objective function

$$\mathcal{L}_\phi^{\text{HC}} = \mathbb{E}_D[(R_t + \gamma(\hat{H}_\phi(\tau_{t_i:t+1}) + \hat{C}_\phi(s_{t+1}, a'_{t+1})) - (H_\phi(\tau_{t_i:t}) + C_\phi(s_t, a_t)))^2] + \lambda L_{\text{reg}}(H_\phi), \tag{8}$$

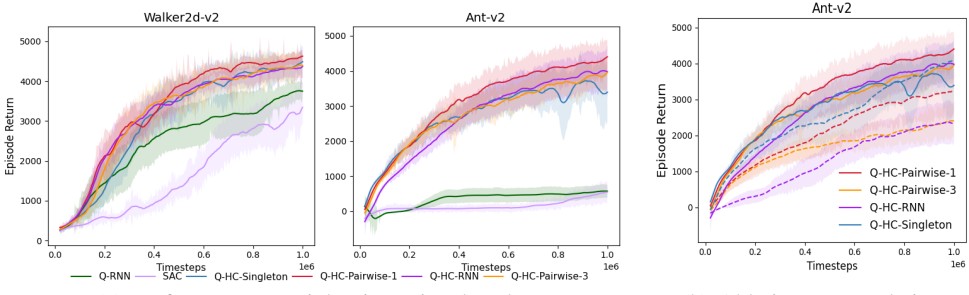

(a) Performance on High-Dimensional Tasks      (b) Ablation on Regulation

Figure 2: **(a)**: Performance of different implementations of HC-decomposition framework ($\mathcal{Q}$-HC), $\mathcal{Q}$-RNN and vanilla SAC. The task is the continuous sum-form PI-DRMDP task and $q_n$ is a uniform distribution between 15 and 20. The results show the mean and standard deviation of 7 seeds each. **(b)** Ablation on the $L_{\text{reg}}$. Each dashed line is the no-regularization version of the corresponding $\mathcal{Q}$-HC of the same color.

where the first term is the TD error in Eq. (4) and the second term is a regularizer on $H_\phi$. We use the regularization to stabilize the optimization process and prevent $H_\phi$ from taking away too much information of the credit on the choice of $a_t$. In Section 4.1, we will compare several simple choices of $H_\phi$ and $L_{\text{reg}}$. The search for other designs of $H_\phi$ and the regularization term in various settings is left as future work.

# 4 EXPERIMENT

In this section, we first test and illustrate our algorithmic framework and HC-decomposition on high-dimensional PI-DRMDP tasks to validate our claims in previous sections. Then, we compare our algorithm with several previous baselines on sum-form delayed reward tasks. Our algorithm turns out to be the SOTA algorithm which is most sample-efficient and stable. Finally, we demonstrate our algorithm via an illustrative example.

## 4.1 DESIGN EVALUATION

We implement the following architectures of $H_\phi$ in our experiment. Our algorithm is based on SAC (Haarnoja et al., 2018a).

$\mathcal{Q}$**-HC-RNN**. $H_\phi(\tau_{t_i:t})$ is approximated with a GRU network.

$\mathcal{Q}$**-HC-Pairwise-K**. $H_\phi(\tau_{t_i:t})$ is the sum of $K+1$ networks $\{c_\phi^k\}_{k=0,1,\dots K}$ for different pairwise terms as follows[2]. $K = 1, 3$ are included in the experiment.

$$H_\phi(\tau_{t_i:t}) = \sum_{k\in[0:K+1]} \sum_{j\in[t_i:t-k]} c_\phi^k(\tau_{j:j+1} \circ \tau_{j+k:j+k+1}).$$

$\mathcal{Q}$**-HC-Singleton**. $H_\phi(\tau_{t_i:t})$ is the sum of single step terms.

$$H_\phi(\tau_{t_i:t}) = \sum_{j\in[t_i:t]} b_\phi(\tau_{j:j+1}).$$

In terms of the representation power of these structures, (-RNN) is larger than (-Pairwise) larger than (-Singleton). With a small representation power, critic's learning will suffer from large projection error. In terms of optimization, (-Singleton) is easier than (-Pairwise) and easier than (-RNN). As discussed in Section 3.2, non-optimized critic will result in inaccurate and unstable policy gradients. The practical design of $H_\phi$ is a trade-off between the efficiency of optimization and the scale of projection error. For the regularizer in Eq. (8), we choose the following for all implementations.

$$L_{\text{reg}}(H_\phi) = \mathbb{E}_D\left[(H_\phi(\tau_i) - r(\tau_i))^2\right], \tag{9}$$

---

[2]Input for $c_\phi^0$ is thus a single state-action pair.

where $\tau_i$ is sampled from the $D$. As discussed in Section 3.2, the history part $H_\phi(\tau_{t_i:t})$ should only be used to infer the credit of $\tau_{t_i:t}$ within the signal interval $\tau_i$. However, without the regularization, the approximated $H_\phi$ may deprive too much information from $\mathcal{Q}_\phi$ than we have expected. Thus, in a direct manner, we regularize $H_\phi(\tau_i)$ to the same value of $r(\tau_i)$.

In Figure 2, we compare these algorithms on sum-form PI-DRMDP continuous control tasks based on OpenAI Gym. Rewards are given once every $n$ steps which is uniformly distributed from 15 to 20. The reward is the sum of the standard per-step reward in these tasks. Our method is compared with vanilla SAC and $\mathcal{Q}$-RNN whose $\mathcal{Q}^\pi$ is approximated by a GRU network. The results verify our claims made in Section 3 and the necessity of the design of our method.

1. Algorithms under the algorithmic framework (i.e., with prefix $\mathcal{Q}$) outperform vanilla SAC. Our framework approximates $\mathcal{Q}^\pi$ while vanilla SAC cannot handle non-Markovian rewards.
2. $\mathcal{Q}^\pi$ approximated with HC-decomposition architecture (i.e., with prefix $\mathcal{Q}$-HC) is more sample-efficient and more stable than $\mathcal{Q}$-RNN. Indeed, the HC architecture results in an easier optimization in critic learning.
3. Figure 2(b) shows the importance of regularization, especially when $H_\phi$ has complex form (i.e., $\mathcal{Q}$-HC-RNN) and thus is more likely to result in ill-optimized $C$-part.

**More Experiments.** In Appendix C.1, we show that the empirical results are consistent in a variety of PI-DRMDP tasks and under different implementations. First, we test in sum-form tasks with different interval length distributions ($q_n$). As shown in Table 2, our $\mathcal{Q}$-HC methods are scalable to sum-form tasks with a signal interval length of 60 environmental steps, which is quite a long period in the Gym benchmark, and perform well in various random length tasks. Then, in non-sum-form tasks with *Max* and *Square* reward functions, we find that $\mathcal{Q}$-HCs still outperform $\mathcal{Q}$-RNN and the vanilla SAC consistently in all environments (Figure 5). This suggests that the advantage of HC-decomposition also holds in other PI reward function tasks. Moreover, in Figure 6, we also test TD3-based $\mathcal{Q}$-HC-Singleton and $\mathcal{Q}$-HC-Pairwise-1. Comparing to TD3-based $\mathcal{Q}$-RNN and vanilla TD3 (Fujimoto et al., 2018), these two variants outperform the baselines uniformly.

To sum up, based on the empirical evidence, we conclude that our $\mathcal{Q}$-HC is widely applicable in PI-DRMDPs as a remedy to the unstable optimization of direct $\mathcal{Q}$ approximation. As $\mathcal{Q}$-HC is an implementation of the general algorithm in Section 3.1, it handles the non-Markovian rewards with theoretical support. In future work, we will explore the boundary of HC-decomposition, and analyze the effect of the projection error mathematically.

## 4.2 COMPARATIVE EVALUATION

Here, we compare our algorithm with previous algorithms in sum-form high-dimensional delayed reward tasks. The following baselines are included in the comparison.

- LIRPG (Zheng et al., 2018). It utilizes intrinsic rewards to make policy gradients more efficient. The intrinsic reward is learnt by meta-gradient from the same objective function. We use the same code provided by the paper.
- RUDDER (Arjona-Medina et al., 2019). It decomposes the delayed and sparse reward to a surrogate per-step reward via regression. Additionally, it utilizes on-policy correction to ensure the same optimality w.r.t the original problem. We alter the code for the locomotion tasks.
- SAC-IRCR (Gangwani et al., 2020). It utilizes off-policy data to provide a smoothed guidance rewards for SAC. As mentioned in the paper, the performance heavily relies on the smoothing policy. We implement a delayed reward version of IRCR in our setting.

For clarity, we only include $\mathcal{Q}$-HC-Pairwise-1 and $\mathcal{Q}$-HC-RNN in the comparison. As shown in Figure 3, we find that LIRPG and RUDDER perform sluggishly in all tasks, due to their on-policy nature (*i.e.*, based on PPO (Schulman et al., 2017)). SAC-IRCR performs well only on some easy tasks (e.g. Hopper-v2). Unfortunately, SAC-IRCR has a bad performance (e.g. Ant-v2, Walker2d-v2, Reacher-v2) on other tasks. We suspect in these tasks, the smoothing technique results in a erroneous guidance reward which biases the agent. Thus, SAC-IRCR is not a safe algorithm in solving delay reward tasks. In contrast, ours (as well as other implementations shown in Appendix C.3) perform well on all tasks and surpass the baselines by a large margin. Most surprisingly, our algorithm can achieve the near optimal performance, i.e., comparing with Oracle SAC which is trained on dense reward environment for 1M steps. Noticing that we use an environment in which rewards are given only every 20 steps.

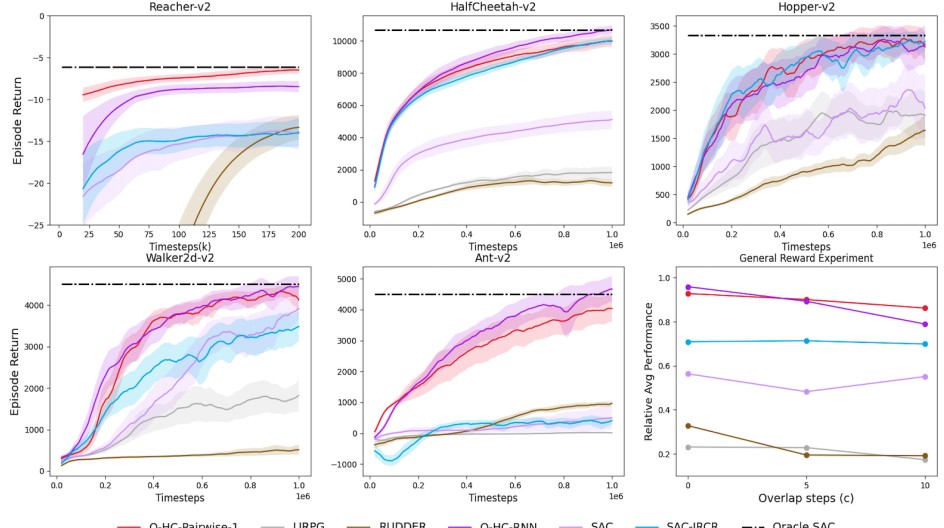

Figure 3: Comparisons of $\mathcal{Q}$-HC-Pairwise-1, $\mathcal{Q}$-HC-RNN and baselines on 5 sum-form PI-DRMDP tasks. The dashed line is the value of Oracle SAC. $q_n$ is fixed as 20 (Zheng et al., 2018; Oh et al., 2018). The results show the mean and the standard deviation of 7 runs. All curves are further smoothed equally for clarity. The last picture shows the relative performance w.r.t the Oracle SAC on sum-form PI DRMDP tasks with general reward function (Appendix B). Each data point shows the average of 5 tasks of each algorithm. X-axis refers to the overlapping steps of the reward function.

We also conduct experiments on tasks with general reward functions, in which our algorithmic framework and the HC-decomposition can be naturally extended to (please refer to Appendix B and Appendix A). In the General Experiment of Figure 3, we plot the relative average performance w.r.t Oracle SAC on tasks whose general reward functions have different amount of overlapped steps ($c = 5, 10$). Clearly, our $\mathcal{Q}$-HC is still the SOTA algorithm for every $c$. Thus, our method is applicable to environments with general reward functions. Please refer to Appendix D for experiment details.

## 4.3 HISTORICAL INFORMATION

In addition, with a toy example, we explore what kind of information the *H*-component learns so that the *C*-component makes suitable decisions. The toy example is a target-reaching task illustrated in Figure 4 Left. The point agent is given delayed reward which roughly indicates its distance to the target area. This mimics the low frequency feedback from the environment. Noticing that the reward function is not in sum-form. Please refer to Appendix D for more details.

The learning curves are shown in Figure 4 Middle. Clearly, $\mathcal{Q}$-HC-Singleton outperforms the baselines by a large margin. For better illustration, we visualize $b_\phi(s_t, a_t)$ in $\mathcal{Q}$-HC-Singleton on the grid in Figure 4 Right. The pattern highlights the line from the start point to the target area (i.e., the optimal policy), suggesting that $b_\phi$ has captured some meaningful patterns to boost training. We also observe a similar pattern for HC-Singleton without regression (Appendix D). This is an interesting discovery which shows that the *H*-component may also possess direct impact on the policy learning instead of simply function as an approximator.

## 5 RELATED WORK

**Off-Policy RL:** Off-policy deep reinforcement learning algorithms TD3 (Fujimoto et al., 2018) and SAC (Haarnoja et al., 2018a) are the most widely accepted actor-critic algorithms on the robot locomotion benchmark (Duan et al., 2016) so far. Based on previous work (Degris et al., 2012; Silver et al., 2014), Fujimoto et al. (2018) puts forward the clipped double Q-learning (CDQ) technique to address overestimation in critic learning. Haarnoja et al. (2018a) also uses CDQ technique but instead optimizes the maximum entropy objective. These two methods lead to a more robust policy.

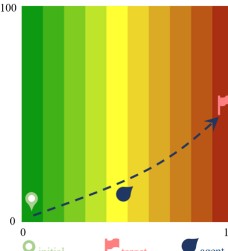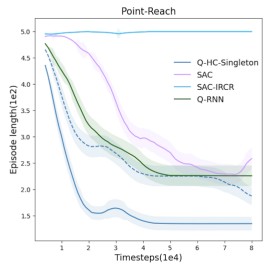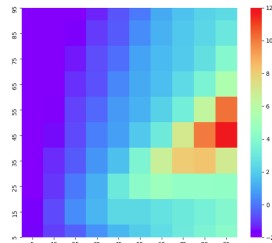

Figure 4: **Left:** $100 \times 100$ Point-Reach task with additional positional reward (indicated by area color). Reward is given every 20 steps. **Middle:** Learning curves of $\mathcal{Q}$-HC-Singleton and several baselines. Dashed line represents $\mathcal{Q}$-HC-Singleton without regularization. Y-axis shows the number of steps needed for the point agent to reach the target area (cut after 500 steps). All curves represent the mean of 10 seeds and are smoothed equally for clarity. **Right:** Heatmap visualization of $b_\phi$ in $\mathcal{Q}$-HC-Singleton.

In our setting, we observe that SAC-based algorithms slightly outperforms TD3-based algorithms suggesting the benefit of maximum entropy in delayed reward tasks.

**Delayed or Episodic Reward RL:** Developing RL algorithms for delayed reward or episodic reward (at the end of the trajectory) has become a popular research area recently. Some theoretical analysis have been conducted on delayed rewards in MAB and MDP Zhou et al. (2018; 2019); Héliou et al. (2020) These tasks are known to be difficult for long-horizon credit assignment (Sutton, 1984). To address this issue, Zheng et al. (2018) proposes to use intrinsic reward to boost the efficiency of policy gradients. The intrinsic reward is learnt via meta-learning (Finn et al., 2020). Gangwani et al. (2019) and Guo et al. (2018) use a discriminator to provide guidance reward for the policy. The discriminator is trained jointly with the policy with binary classification loss for self-imitating. In RUDDER (Arjona-Medina et al., 2019), it utilizes a recurrent network to predict a surrogate per-step reward for guidance. To ensure the same optimality, the guidance is then corrected with ground truth reward. Liu et al. (2019) extends the design and utilizes a Transformer (Vaswani et al., 2017) network for better credit assignment on episodic reward tasks. Recently, Klissarov & Precup (2020) proposes to use GCN (Kipf & Welling, 2016) network to learn a potential function for reward shaping (Ng et al., 1999). Noticing that all these algorithms are based on PPO (Schulman et al., 2017) and thus are on-policy algorithms while ours is an off-policy one.

Most recently, Gangwani et al. (2020) proposes to augment off-policy algorithms with a trajectory-space smoothed reward in episodic reward setting. The design turns out to be effective in solving the problem. However, as mentioned by the paper itself, this technique lacks theoretical guarantee and heavily relies the choice of the smoothing policy. As shown in Section 4.2, in many cases, this method becomes extremely spurious. Our algorithm is derived from a theoretical perspective and performs well on all tasks.

## 6 CONCLUSION

In this paper, we model the sequential decision problem with delayed rewards as Past-Invariant Delayed Reward MDPs. As previous off-policy RL algorithms suffer from multiple problems in PI-DRMDP, we put forward a novel and general algorithmic framework to solve the PI-DRMDP problems that has theoretical guarantees in the tabular case. The framework relies on a novelly defined $\mathcal{Q}$-value. However, in high dimensional tasks, it is hard to approximate the $\mathcal{Q}$-value directly. To address this issue, we propose to use the HC-approximation framework for stable and efficient training in practice. In the experiment, we compare different implementations of the HC framework. They all perform well and robustly in continuous control PI-DRMDP locomotion tasks based on OpenAI Gym. Besides, our method outperforms previous baselines on delayed reward tasks remarkably, suggesting that our algorithm is a SOTA algorithm on these tasks so far.

In terms of future work, two research directions are worth exploring. One is to develop our algorithm and the HC-approximation scheme to various real world settings mentioned in Section 1, possibly with offline data (Levine et al., 2020). The other direction is to design efficient and advanced algorithms with theoretical guarantees for the general DRMDP tasks.

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

## A  Algorithm

We summarize our algorithm for PI-DRMDP with General Reward Function as follows.

---

**Algorithm 1** Algorithm with General Reward Function

---

1: Choose $\mathcal{Q}_\phi$ structure: $\mathcal{Q}$-RNN, $\mathcal{Q}$-HC-Pairwise-K, e.t.c
2: Initialize $\pi_\theta$, $\mathcal{Q}_\phi$ and target networks, $D \leftarrow \emptyset$
3: **for** each env step **do**
4:     $a_t \sim \pi_\theta(\cdot|s_t, t - t_i)$
5:     Take $a_t$ and get $s_{t+1}, R_t$
6:     Add $(s_t, a_t, R_t, s_{t+1})$ to $D$
7:     **for** each gradient step **do**
8:         Sample a batch of $(\tau_{t_i-c:t+1}, R_t, s_{t+1})$ from $D$
9:         Update $\mathcal{Q}_\phi$ via minimizing Eq. (8) (with $\tau_{t_i-c:t}$) for $\mathcal{Q}$-HC or Eq. (11) for $\mathcal{Q}$-RNN
10:         Update $\pi_\theta$ via Eq. (7) for $\mathcal{Q}$-HC or Eq. (12) for $\mathcal{Q}$-RNN
11:         Update the target networks
12:     **end for**
13: **end for**

---

## B  Discussions and Formal Proofs

In this part, we provide the proofs for the statements in the main paper. In addition, we also add detailed discussions on the issues mentioned above. To begin with, we restate the definitions with general reward functions.

**Definition 3** (DRMDP with General Reward Function, DRMDP-c). *A DRMDP-c $M_c = (S, A, p, q_n, r_c, \gamma)$ is described by the following parameters.*

*1. The state and action spaces are $S$ and $A$ respectively.*

*2. The Markov transition function is $p(s'|s, a)$ for each $(s, a) \in S \times A$; the initial state distribution is $p(s_0)$.*

*3. The signal interval length is distributed according to $q_n(\cdot)$, i.e., for the $i$-th signal interval, its length $n_i$ is independently drawn from $q_n(\cdot)$.*

*4. The general reward function $r_c$ defines the expected reward generated for each signal interval with a overlap of maximal $c$ steps with previous signal intervals; suppose $\tau_i = \tau_{t:t+n_i} = (s, a)_{t:t+n_i} = ((s_t, a_t), \ldots (s_{t+n-1}, a_{t+n_i-1}))$ is the state-action sequence during the $i$-th signal interval of length $n_i$, then the expected reward for this interval is $r_c(\tau_{t-c:t+n_i})$. [3]*

*5. The reward discount factor is $\gamma$.*

The PI-DRMDP is also extended naturally as follows.

**Definition 4** (PI-DRMDP with General Reward Function, PI-DRMDP-c). *A Past-Invariant DRMDP-c is a DRMDP-c $M_c = (S, A, p, q_n, r_c, \gamma)$ whose reward function $r_c$ satisfies the following Past-Invariant (PI) condition: for any two trajectory segments $\tau_1$ and $\tau_2$ of the same length (no less than $c$), and for any two equal-length trajectory segments $\tau_1'$ and $\tau_2'$ such that the concatenated trajectories $\tau_a \circ \tau_b'$ are feasible under the transition dynamics $p$ for all $a, b \in \{1, 2\}$, it holds that*

$$r(\tau_1 \circ \tau_1') > r(\tau_1 \circ \tau_2') \iff r(\tau_2 \circ \tau_1') > r(\tau_2 \circ \tau_2').$$

**Remark 1.** *Definition 1 and Definition 2 are the special case ($c = 0$) of Definition 3 and Definition 4 respectively.*

Clearly, we have the following Fact.

**Fact 3.** $\forall c' < c$, *DRMDP $M_{c'}$ is also a DRMDP $M_c$ and PI-DRMDP $M_{c'}$ is also a PI-DRMDP $M_c$.*

---

[3]If $t < c$, $\tau_{t-c:0}$ refers to some zero token paddings which indicates the beginning of the trajectory.

Besides, under the general reward function, we extend the $\mathcal{Q}$ definition.

$$\mathcal{Q}^\pi(\tau_{t_i-c:t+1}) := \mathbb{E}_{(\tau,n)\sim\pi}\left[\sum_{j=i}^\infty \gamma^{t_{j+1}-t-1}r(\tau_{t_j-c:t_{j+1}})\Big|\tau_{t_i-c:t+1}\right]. \tag{10}$$

Correspondingly, $\mathcal{Q}^\pi(\tau_{t_i-c:t+1})$ is optimized via

$$\mathcal{L}_\phi := \mathbb{E}_D\left[(R_t + \gamma\hat{\mathcal{Q}}_\phi(\tau_{t_j-c:t+2}) - \mathcal{Q}_\phi(\tau_{t_i-c:t+1}))^2\right], \tag{11}$$

where $\tau_{t_j-c:t+2} = \tau_{t_i-c:t+1} \circ (s_{t+1}, a'_{t+1})$ (so that $j = i$) if $t$ is not the last step of $\tau_i$, and $\tau_{t_j-c:t+2} = \tau_{t_{i+1}-c:t_{i+1}} \circ (s_{t_{i+1}}, a'_{t_{i+1}})$ (so that $j = i+1$) otherwise.

The general policy update is also extended to

$$\nabla_\theta\mathcal{J}(\pi_\theta) = \mathbb{E}_D\left[\nabla_{a_t}\mathcal{Q}^\pi(\tau_{t_i-c:t}\circ(s_t,a_t))\Big|_{\pi_\theta(s_t,t-t_i)}\nabla_\theta\pi_\theta(s_t,t-t_i)\right]. \tag{12}$$

**Without specification, the following proofs hold for $\forall c$.**

### B.1    PROOF OF FACT 1

**Fact 4** (Fact 1 with General Reward Function). *For any distribution $D$ with non-zero measure for any $\tau_{t_i-c:t+1}$, $\mathcal{Q}^\pi(\tau_{t_i-c:t+1})$ is the unique fixed point of the MSE problem in Eq. (4). More specifically, when fixing $\hat{\mathcal{Q}}_\phi$ as the corresponding $\mathcal{Q}^\pi$, the solution of the MSE problem is still $\mathcal{Q}^\pi$.*

*Proof.* Though we state the fact for PI-DRMDP-c, we will prove it for general $\pi \in \Pi_{\tau,c}$ in DRMDP-c setting. For brevity, we replace $t+1$ with $t$ in the proof. By definition in Eq. (10), we have

$$\mathcal{Q}^\pi(\tau_{t_i-c:t}) = q_n(n_i \neq t - t_i | n_i \geq t - t_i)$$

$$\left(\gamma\sum_{s_t,a_t}p(s_t|s_{t-1},a_{t-1})\pi(a_t|\tau_{t_i-c:t}\circ s_t)\mathcal{Q}^\pi(\tau_{t_i-c:t}\circ(s_t,a_t))\right)$$

$$+ q_n(n_i = t - t_i | n_i \geq t - t_i)$$

$$\left(r(\tau_{t_i-c:t}) + \gamma\sum_{s_t,a_t}p(s_t|s_{t-1},a_{t-1})\pi(a_t|\tau_{t-c:t}\circ s_t)\mathcal{Q}^\pi(\tau_{t-c:t}\circ(s_t,a_t))\right). \tag{13}$$

Clearly, $\mathcal{Q}^\pi(\tau_{t_i-c:t})$ is the solution of the following MSE problem. Noticing that we assume $\mathcal{Q}_\phi$ has tabular expression over $\tau_{t_i-c:t}$s.

$$\min_\phi \sum_{R_{t-1},s_t} p_D(\tau_{t_i-c:t}, R_{t-1}, s_t)\pi(a'_t|\tau_{t_j-c:t}\circ s_t)\left(R_{t-1} + \gamma\mathcal{Q}^\pi(\tau_{t_j-c:t}\circ(s_t,a'_t)) - \mathcal{Q}_\phi(\tau_{t_i-c:t})\right)^2$$

Here, $\tau_{t_j-c:t}$ is defined similarly as in the main text and $p_D$ refers to the probability of the fragment of trajectory is sampled from the buffer $D$. We denote the objective function as $\mathcal{L}_\phi(\tau_{t_i-c:t})$. As $\mathcal{L}_\phi$ in Eq. (11) is the sum of all $\mathcal{L}_\phi(\tau_{t_i-c:t})$, $\mathcal{Q}^\pi(\tau_{t_i-c:t})$ is a fixed point of the MSE problem.

The uniqueness is proved similar to the standard MDPs setting. For any $\hat{\mathcal{Q}}_\phi$, we denote the optimal solution of Eq. (4) as $\mathcal{Q}_\phi(\tau_{t_i-c:t})$. Since we assume tabular expression and non-zero probability of any $p_D(\tau_{t_i-c:t})$ Then, $\forall\tau_{t_i-c:t}$, we have

$$\left|\mathcal{Q}_\phi(\tau_{t_i-c:t}) - \mathcal{Q}^\pi(\tau_{t_i-c:t})\right| \leq \gamma q_n(n_i \neq t - t_i | n_i \geq t - t_i)\sum_{s_t,a_t}p(s_t|s_{t-1},a_{t-1})\pi(a_t|\tau_{t_i-c:t}\circ s_t)$$

$$\left|\hat{\mathcal{Q}}_\phi(\tau_{t_i-c:t+1}) - \mathcal{Q}^\pi(\tau_{t_i-c:t+1})\right| + \gamma q_n(n_i = t - t_i | n_i \geq t - t_i)$$

$$\sum_{s_t,a_t}p(s_t|s_{t-1},a_{t-1})\pi(a_t|\tau_{t-c:t}\circ s_t)$$

$$\left|\hat{\mathcal{Q}}_\phi(\tau_{t-c:t}\circ(s_t,a_t)) - \mathcal{Q}^\pi(\tau_{t-c:t}\circ(s_t,a_t))\right|$$

$$\leq \gamma\parallel\hat{\mathcal{Q}}_\phi - \mathcal{Q}^\pi\parallel_\infty$$

Last term denotes the infinite norm of the vector of residual value between $\mathcal{Q}^\pi$ and $\hat{\mathcal{Q}}_\phi$. Each entry corresponds to some $\tau_{t_i - c:t}$ feasible in the dynamic. Consequently, by the $\gamma-$concentration property, if $\mathcal{Q}_\phi = \hat{\mathcal{Q}}_\phi$ after the iteration (*i.e.*, $\mathcal{Q}_\phi$ is a fixed point), then $\mathcal{Q}_\phi = \mathcal{Q}^\pi$. This completes the proof of uniqueness. $\qquad\square$

**Remark 2.** *Fact 1 holds as the special case of Fact 4 when $c = 0$.*

### B.2    PROOF OF FACT 2

**Fact 5** (Fact 2 with General Reward Function). *For any $\tau_{t_i - c:t}, \tau'_{t_i - c:t}$ and $s_t, a_t, a'_t$ which both $\tau_{t_i - c:t} \circ s_t$ and $\tau'_{t_i - c:t} \circ s_t$ are feasible under the transition dynamics, $\forall \pi \in \Pi_s$, we have that*

$$\mathcal{Q}^\pi(\tau_{t_i - c:t} \circ (s_t, a_t)) > \mathcal{Q}^\pi(\tau_{t_i - c:t} \circ (s_t, a'_t)) \iff \mathcal{Q}^\pi(\tau'_{t_i - c:t} \circ (s_t, a_t)) > \mathcal{Q}^\pi(\tau'_{t_i - c:t} \circ (s_t, a'_t))$$

*Proof.* Since $\pi(a_t | s_t, t - t_i) \in \Pi_s$, for any $s_t, a_t$, the distribution of the remaining trajectories $\rho^\pi(\tau_{t:T}, n_{t:T} | s_t, a_t, t_i)$ is irrelevant with $\tau_{t_i - c:t}$ . Thus, we have

$$
\begin{aligned}
\mathcal{Q}^\pi(\tau_{t_i - c:t} \circ (s_t, a_t)) &= \sum_{\tau_{t:\infty}, n_{t:\infty}} \rho^\pi(\tau_{t:\infty}, n_{t:\infty} | s_t, a_t, t_i) \left( \sum_{j=i}^{\infty} \gamma^{t_{j+1} - t - 1} r(\tau_{t_j - c:t_{j+1}}) \right) \\
&= \sum_{t_{i+1}} \gamma^{t_{i+1} - t - 1} q_n(n_i = t_{i+1} - t_i | n_i \geq t - t_i) \sum_{\tau_{t:t_{i+1}}} \rho^\pi(\tau_{t:t_{i+1}} \circ s_{t_{i+1}} | s_t, a_t, t_i) \\
&\quad \cdot \left( r(\tau_{t_i - c:t_{i+1}}) + \gamma \sum_{a_{t_{i+1}}} \pi(a_{t_{i+1}} | s_{t_{i+1}}, 0) \mathcal{Q}^\pi(\tau_{t_{i+1} - c:t_{i+1}} \circ (s_{t_{i+1}}, a_{t_{i+1}})) \right) \\
&\propto \sum_{t_{i+1}, \tau_{t:t_{i+1}}} q_n(n_i = t_{i+1} - t_i | n_i \geq t - t_i) \rho^\pi(\tau_{t:t_{i+1}} | s_t, a_t, t_i) r(\tau_{t_i - c:t_{i+1}}) \\
&\quad + K(s_t, a_t, t_i)
\end{aligned}
$$

The last term denotes the residual part which is irrelevant with $\tau_{t_i:t}$. Besides, under the PI condition, the order of $r(\tau_{t_i - c:t_{i+1}})$ is invariant over $\tau_{t_i - c:t}$. As a result, the order of $\mathcal{Q}^\pi(\tau_{t_i - c:t} \circ (s_t, a))$ among $a$ is also invariant of $\tau_{t_i - c:t}$. This completes the proof. $\qquad\square$

**Remark 3.** *Fact 2 holds as then special case of Fact 5 when $c = 0$.*

### B.3    PROOF OF PROPOSITION 1

**Proposition 2** (Proposition 1 with General Reward Function). *For any policy $\pi_k \in \Pi_s$, the policy iteration w.r.t. $\mathcal{Q}^{\pi_k}$ (Eq. (10)) produces policy $\pi_{k+1} \in \Pi_s$ such that $\forall \tau_{t_i - c:t+1}$, it holds that*

$$\mathcal{Q}^{\pi_{k+1}}(\tau_{t_i - c:t+1}) \geq \mathcal{Q}^{\pi_k}(\tau_{t_i - c:t+1}),$$

*which implies that $\mathcal{J}(\pi_{k+1}) \geq \mathcal{J}(\pi_k)$.*

*Proof.* Under Fact 5, the new policy iteration is defined formally as following

$$\pi_{k+1}(a_t | s_t, t - t_i) = \arg\max_{a_t} \mathcal{Q}^{\pi_k}(\tau_{t-c:t} \circ (s_t, a_t))$$

where $\tau_{t-c:t}$ is any trajectory segments feasible for $s_t$ under the dynamics. If there are multiple maximums, we can assign arbitrary probability among these $a_t$s. Following Eq. (13), we have

$\forall \tau_{t_i-c:t}$

$$
\begin{aligned}
\mathcal{Q}^{\pi_k}(\tau_{t_i-c:t}) =\ & q_n(n_i \neq t-t_i|n_i \geq t-t_i) \\
& \left( \gamma \sum_{s_t,a_t} p(s_t|s_{t-1},a_{t-1})\pi_k(a_t|s_t,t-t_i)\mathcal{Q}^{\pi_k}(\tau_{t_i-c:t}\circ(s_t,a_t)) \right) \\
& q_n(n_i = t-t_i|n_i \geq t-t_i) \\
& \left( r(\tau_{t_i-c:t}) + \gamma \sum_{s_t,a_t} p(s_t|s_{t-1},a_{t-1})\pi_k(a_t|s_t,0)\mathcal{Q}^{\pi_k}(\tau_{t-c:t}\circ(s_t,a_t)) \right) \\
\leq\ & q_n(n_i \neq t-t_i|n_i \geq t-t_i) \\
& \left( \gamma \sum_{s_t,a_t} p(s_t|s_{t-1},a_{t-1})\pi_{k+1}(a_t|s_t,t-t_i)\mathcal{Q}^{\pi_k}(\tau_{t_i-c:t}\circ(s_t,a_t)) \right) + \\
& q_n(n_i = t-t_i|n_i \geq t-t_i) \\
& \left( r(\tau_{t_i-c:t}) + \gamma \sum_{s_t,a_t} p(s_t|s_{t-1},a_{t-1})\pi_{k+1}(a_t|s_t,0)\mathcal{Q}^{\pi_k}(\tau_{t-c:t}\circ(s_t,a_t)) \right) \quad (14)
\end{aligned}
$$

Then, we can iteratively extend each $\mathcal{Q}^{\pi_k}$ term in Eq. (14) and get $\forall \tau_{t_i-c:t}$

$$
\mathcal{Q}^{\pi_k}(\tau_{t_i-c:t}) \leq \mathcal{Q}^{\pi_{k+1}}(\tau_{t_i-c:t})
$$

Noticing that the inequality is strict if any step of the inequality in the iterations is strict. Furthermore, by definition, $\mathcal{J}(\pi) = \sum_{s_0,a_0} p(s_0)\pi(a_0|s_0,0)\mathcal{Q}^{\pi}(s_0,a_0)$[4], we have

$$
\begin{aligned}
\mathcal{J}(\pi_k) &\leq \sum_{s_0,a_0} p(s_0)\pi_{k+1}(a_0|s_0,0)\mathcal{Q}^{\pi_k}(s_0,a_0) \\
&\leq \sum_{s_0,a_0} p(s_0)\pi_{k+1}(a_0|s_0,0)\mathcal{Q}^{\pi_{k+1}}(s_0,a_0) \\
&\leq \mathcal{J}(\pi_{k+1})
\end{aligned}
$$

This completes the proof. $\qquad\square$

**Remark 4.** *Proposition 1 holds as the special case of Proposition 2 when $c = 0$.*

### B.4 DISCUSSION OF OPTIMAL POLICIES

To begin with, we consider the general DRMDP-c. Similarly, we denote $\Pi_{\tau,c} = \{\pi(a_t|\tau_{t_i-c:t}\circ s_t)\}$ as the extension of $\Pi_\tau$ ($c = 0$). Then, we prove the following Fact.

**Fact 6.** *For any DRMDP-c, there exists an optimal policy $\pi^* \in \Pi_{\tau,c}$. However, there exists some DRMDP-0 (so as DRMDP-c $\forall c$) such that all of its optimal policies $\pi^* \notin \Pi_s$.*

In general, any policy belongs to the class $\Pi_{\text{all}} = \{\pi(a_t|\tau_{0:t}\circ s_t, n_{0:t})\}$. Namely, the agent's policy can only base on all the information it has experienced till step $t$. Besides, in DRMDP-c, the expected discounted cumulative reward is

$$
\mathcal{J}(\pi) = \sum_{\tau,n} \rho^{\pi}(\tau,n)R(\tau,n).
$$

where $\rho^{\pi}(\tau,n)$ denotes the probability that trajectory $(\tau,n)$ is generated under policy $\pi \in \Pi_{\text{all}}$. To begin with, we first prove the following Lemma.

**Lemma 1.** *For any policy $\pi \in \Pi_{\text{all}}$ which is the optimal, i.e, maximize $\mathcal{J}(\pi)$, we consider two segments of trajectories $(\tau^a_{0:t^a}, n^a_{0:t^a})$ and $(\tau^b_{0:t^b}, n^b_{0:t^b})$ which satisfies that $\tau^a_{t_i-c:t^a} = \tau^b_{t_i-c:t^b}$ and $t^b \geq t^a$. Besides, we also consider $\forall t \geq t^b, t' = t - t^b + t^a$ and $\tau^b_{0:t} = \tau^b_{0:t^b}\circ\tau_{t^b:t}, n^b_{0:t} = n^b_{0:t^b}\circ n_{t^b:t}$*

---

[4] Here, we omit specifying the $c$-step padding.

and state $s_t^{(')}$ feasible in the dynamic, we switch the policy $\pi$ to $\pi'$ w.r.t these two trajectories as following

$$\pi'(\cdot|\tau_{0:t}^b \circ s_t, n_{0:t}^b) = \pi(\cdot|\tau_{0:t^a}^a \circ \tau_{t^b:t} \circ s_{t'}, n_{0:t^a}^a \circ n_{t^b:t}).$$

Then, $\pi' \in \Pi_{all}$ and $\mathcal{J}(\pi') = \mathcal{J}(\pi)$.

*Proof.* Obviously, $\pi'$ is well-defined and $\pi' \in \Pi_{all}$ as the transition is Markovian. Noticing that

$$\mathcal{J}(\pi') = \sum_{\tau,n} \rho^{\pi'}(\tau,n) R(\tau,n)$$

$$= \underbrace{\sum_{(\tau,n):(\tau_{0:t^b}^b, n_{0:t^b}^b) \not\subset (\tau,n)} \rho^{\pi'}(\tau,n) R(\tau,n)}_{\text{denoted by } J^{\pi'}(\tau_{0:t^b}^b, n_{0:t^b}^b)} + \rho^{\pi'}(\tau_{0:t^b}^b, n_{0:t^b}^b)$$

$$\sum_{(\tau,n):(\tau_{0:t^b}^b, n_{0:t^b}^b) \subset (\tau,n)} \rho^{\pi'}(\tau,n|\tau_{0:t^b}^b, n_{0:t^b}^b) R(\tau,n)$$

$$= J^{\pi'}(\tau_{0:t^b}^b, n_{0:t^b}^b) + \rho^{\pi'}(\tau_{0:t^b}^b, n_{0:t^b}^b) \sum_{(\tau,n):(\tau_{0:t^b}^b, n_{0:t^b}^b) \subset (\tau,n)}$$

$$\rho^{\pi'}(\tau,n|\tau_{0:t^b}^b, n_{0:t^b}^b) \left( \sum_{j=1}^{\infty} \gamma^{t_{j+1}-1} r(\tau_{t_j-c:t_{j+1}}) \right)$$

$$= J^{\pi'}(\tau_{0:t^b}^b, n_{0:t^b}^b) + \rho^{\pi'}(\tau_{0:t^b}^b, n_{0:t^b}^b) \left( \sum_{j=1}^{i(t^b)-1} \gamma^{t_{j+1}-1} r(\tau_{t_j-c:t_{j+1}}) \right)$$

$$+ \rho^{\pi'}(\tau_{0:t^b}^b, n_{0:t^b}^b) \left( \gamma^{t^b-1} \mathcal{Q}^{\pi'}(\tau_{t_i^b-c:t^b}^b) \right).$$

We denote $(\tau_{0:t}, n_{0:t}) \subset (\tau,n)$ if the former is the trajectory prefix of the latter. $\mathcal{Q}^{\pi'}$ is defined in Eq. (3) except that $\pi' \in \Pi_{all}$ in this case. By the definition of $\pi'$ and the condition that $\pi$ is optimal, we have $\mathcal{Q}^{\pi'}(\tau_{t_i^b-c:t^b}^b) = \mathcal{Q}^{\pi}(\tau_{t_i^a-c:t^a}^a) = \mathcal{Q}^{\pi}(\tau_{t_i^b-c:t^b}^b)$ and $J^{\pi'}(\tau_{0:t^b}^b, n_{0:t^b}^b) = J^{\pi}(\tau_{0:t^b}^b, n_{0:t^b}^b)$. Thus, we have

$$\mathcal{J}(\pi') = J^{\pi'}(\tau_{0:t^b}^b, n_{0:t^b}^b) + \rho^{\pi'}(\tau_{0:t^b}^b, n_{0:t^b}^b) \left( \sum_{j=1}^{i(t^b)-1} \gamma^{t_{j+1}-1} r(\tau_{t_j-c:t_{j+1}}) \right)$$

$$+ \rho^{\pi'}(\tau_{0:t^b}^b, n_{0:t^b}^b) \left( \gamma^{t^b-1} \mathcal{Q}^{\pi}(\tau_{t_i^b-c:t^b}^b) \right)$$

$$= J^{\pi}(\tau_{0:t^b}^b, n_{0:t^b}^b) + \rho^{\pi}(\tau_{0:t^b}^b, n_{0:t^b}^b) \sum_{(\tau,n):(\tau_{0:t^b}^b, n_{0:t^b}^b) \subset (\tau,n)} \rho^{\pi}(\tau,n|\tau_{0:t^b}^b, n_{0:t^b}^b)$$

$$\left( \sum_{j=1}^{\infty} \gamma^{t_{j+1}-1} r(\tau_{t_j-c:t_{j+1}}) \right)$$

$$= \mathcal{J}(\pi).$$

$\square$

We refer it as a **policy switch on** $\pi$ **w.r.t** $(\tau_{0:t^b}^b, n_{0:t^b}^b)$ **and** $(\tau_{0:t^a}^a, n_{0:t^a}^a)$ **to** $\pi'$. Then, we prove Fact 6 as follows.

*Proof of Fact 6.* For simplicity, we denote $\Pi_{\tau,c}^t \subset \Pi_{all}$ as the policy space that if $\pi \in \Pi_{\tau,c}^t$ and $\forall t_a, t_b \leq t$ then $\forall (\tau_{0:t^a}^a, n_{0:t^a}^a), (\tau_{0:t^b}^b, n_{0:t^b}^b)$ which $\tau_{t_i^a-c:t^a}^a = \tau_{t_i^b-c:t^b}^b$, $\pi(\cdot|\tau_{0:t^a}^a \circ s_{t^a}, n_{0:t^a}^a) = \pi(\cdot|\tau_{0:t^b}^b \circ s_{t^b}, n_{0:t^b}^b)$ for all $s$ feasible. Clearly, $\Pi_{\tau,c}^0 = \Pi_{all}$ and $\Pi_{\tau,c}^\infty = \Pi_{\tau,c}$. Then, starting from some optimal policy $\pi_0 \in \Pi_{\tau,c}^0$, we utilize the policy switch operation to shift it into $\Pi_\tau$.

Suppose that $\pi_{t-1} \in \Pi_{\tau,c}^{t-1}$, we consider the following two steps of operations.

1. **Step 1**. $\forall (\tau_{0:t}, n_{0:t})$ that $\exists (\tau'_{0:t'}, n'_{t'_i:t'})$ which $t' < t$ and $\tau_{t_i-c:t} = \tau'_{t'_i-c:t'}$ (randomly choose one if multiple segments exist), we conduct a policy switch on $\pi_t$ w.r.t $(\tau_{0:t}, n_{0:t})$ and $(\tau_{0:t'}, n_{0:t'})$. We denote the policy after all these policy switches as $\pi'_{t-1}$.

2. **Step 2**. We denote $S_{\pi_{t-1}}(\tilde{\tau}_{t_i-c:t}) = \{(\tau_{0:t}, n_{0:t}) | \tau_{t_i-c:t} = \tilde{\tau}_{t_i-c:t}$ and $\nexists (\tau'_{0:t'}, n'_{0:t'}), t' < t$ which $\tau'_{t'_i-c:t'} = \tilde{\tau}_{t_i-c:t}\}$. For each $S_{\pi_{t-1}}(\tilde{\tau}_{t_i-c:t})$, we randomly choose some $(\tau^0_{0:t}, n^0_{0:t}) \in S_{\pi_{t-1}}(\tilde{\tau}_{t_i-c:t})$ and conduct policy switch on $\pi'_{t-1}$ w.r.t all $(\tau, n) \in S_{\pi_{t-1}}(\tilde{\tau}_{t_i-c:t})$ and $(\tau^0_{0:t}, n^0_{0:t})$. Finally, we get $\pi_t$.

Since $\pi_{t-1} \in \Pi_{\tau,c}^{t-1}$, it is straightforward to check that $\pi_t \in \Pi_{\tau,c}^t$. Besides, as all operations are policy switches, $\pi_t$ is optimal if $\pi_{t-1}$ is optimal. Consequently, by induction from the optimal policy $\pi_0$, we can prove that $\exists \pi_\infty \in \Pi_{\tau,c}$ which is optimal.

For the second statement, we consider the following DRMDP-0.

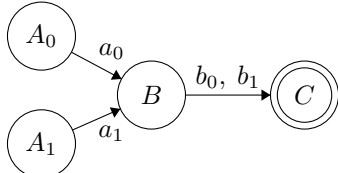

whose $n = 2$. $a_0, a_1$ denote two actions from $A_0, A_1$ to $B$ respectively and $b_0, b_1$ are two actions from B to C. The reward is defined as $r(a_i, b_j) = i \oplus j, \forall (i, j) \in \{0, 1\}$. The initial state distribution is $p(A_0) = p(A_1) = 0.5$. Clearly, for any optimal policy, the agent has to query for its action from the initial state to B before deciding the action from B to C. This illustrates that $\pi^*$ may not be in $\Pi_s$. □

Consequently, learning optimal policies is relatively hard in DRMDP due to the large policy class to search. In our work on PI-DRMDP, we only focus on the optimization in $\Pi_s$. However, in some bizarre cases, the optimal policy $\pi^* \notin \Pi_s$. For example, consider the following PI-DRMDP-0 where $n$ if fixed as 2.

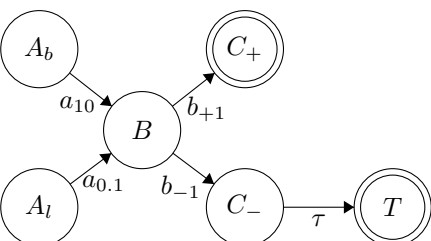

The initial state is uniformly chosen from $\{A_b, A_l\}$ and $C_+, T$ are terminal states. The reward function for the first interval is $r(a_i, b_j) = i \times j$ (Past-Invariant) and $r(\tau) = 5$. The optimal policy at B should be $\pi(\cdot | A_{b/l}, B) = b_{+1/-1}$ which is *not* in $\Pi_s$.

We have not covered the potential non-optimal issue in this work and leave developing general algorithms for acquiring optimal policy for any PI-DRMDP-$c$ or even DRMDP-$c$ as future work. As mentioned above, most real world reward functions still satisfy that the optimal policy belongs to $\Pi_s$. This makes our discussion in $\Pi_s$ still practically meaningful. Additionally, we put forward a sub-class of PI-DRMDP-c which guarantees that there exists some optimal policy $\pi^* \in \Pi_s$.

**Definition 5** (Strong PI-DRMDP-c). *A Strong Past-Invariant DRMDP-c is a PI-DRMDP-c whose reward $r$ satisfies a stronger condition than the Past-Invariant (PI) condition:*
*$\forall$ trajectory segments $\tau_1$ and $\tau_2$ of the same length (no less than c), and for any two equal-length trajectory segments $\tau'_1$ and $\tau'_2$ such that the concatenated trajectories $\tau_a \circ \tau'_b$ are feasible under the transition dynamics $p$ for all $a, b \in \{1, 2\}$, it holds that*

$$r(\tau_1 \circ \tau'_1) - r(\tau_1 \circ \tau'_2) = r(\tau_2 \circ \tau'_1) - r(\tau_2 \circ \tau'_2).$$

Namely, the discrepancies are invariant of the past history $\tau_{1,2}$ in addition to only the order. A straightforward example is the weighted linear sum of per-step rewards

$$r(\tau_{t-c:t+n}) = \sum_{i=t-c}^{t+n-1} w_{i-t} r(s_i, a_i), \quad 0 \le w_{i-t} \le 1$$

The weight mimics the scenario that the per-step reward for the $i-t^{\text{th}}$ step in the reward interval will be lost with some probability $w_{i-t}$. Furthermore, Strong PI-DRMDP-c satisfies

**Fact 7.** *For any Strong PI-DRMDP-c, there exists some optimal policy $\pi^* \in \Pi_s$.*

**Remark 5.** *Table 1 summarizes the case when $c = 0$.*

### B.5 DISCUSSION OF OFF-POLICY BIAS

This following example illustrates the off-policy bias concretely. We consider the case for a sum-form PI-DRMDP-0.

The signal interval length is fixed as $n + 1$ and any trajectory ends after one interval. The reward function is in sum-form over $r(s, a)$. When learning the critic (tabular expression) via Eq. (1), the $Q$-values at the last step $n$ are [5]

$$Q_\phi(s_n, a_n) = \sum_{\tau_{0:n}} \rho^{\boldsymbol{\beta}}(\tau_{0:n}|s_n, a_n) r(\tau_1)$$

$$= \underbrace{\sum_{\tau_{0:n}} \rho^{\boldsymbol{\beta}}(\tau_{0:n}|s_n, a_n) \left( \sum_{t=0}^{n-1} r(\tau_{t:t+1}) \right)}_{\widetilde{Q^{\boldsymbol{\beta}}}(s_n, a_n)} + r(s_n, a_n) \tag{15}$$

In Eq. (15), $\rho^{\boldsymbol{\beta}}(\tau_{0:n}|s_n, a_n)$ denotes the probability distribution of $\tau_{0:n}$ collected under the policy sequence $\boldsymbol{\beta}$ conditioning on that $\tau_{n:n+1} = (s_n, a_n)$. Besides, we denote $\rho^{\beta_k}(\tau_{0:n}|s_n)$ as the distribution under policy $\beta_k$ conditioning on $s_n$ at the last step. Obviously, $\rho^{\beta_k}$ is independent of last step's action $a_n$ as $\pi \in \Pi_s$. By Bayes rule, we have the following relation

$$\rho^{\boldsymbol{\beta}}(\tau_{0:n}|s_n, a_n) = \frac{\sum_{k=1}^{K} \rho^{\beta_k}(\tau_{0:n}|s_n) \beta_k(a_n|s_n)}{\sum_{k=1}^{K} \beta_k(a_n|s_n)} \tag{16}$$

In other word, $\rho^{\boldsymbol{\beta}}(\cdot|s_n, a_n)$ is a weighted sum of $\{\rho^{\beta_k}(\cdot|s_n)\}$. The weights are different between $a_n$s since the behavior policies $\{\beta_k\}$ are different. Consequently, $\widetilde{Q^{\boldsymbol{\beta}}}(s_n, a_n)$ will also vary between $a_n$s and it is hard to be quantified in practice. Unfortunately, we may want $Q_\phi(s_n, a_n) \propto r(s_n, a_n)$ for unbiased credit assignment the actions at the last step. Thus, updating the policy with approximated $Q_\phi(s_n, a_n)$ will suffer from the off-policy bias term $\widetilde{Q^{\boldsymbol{\beta}}}(s_n, a_n)$.

### B.6 DISCUSSION OF FIXED POINT BIAS

Even with on-policy samples in $D$, critic learning via Eq. (1) can still be erroneous. We formally state the observation via a toy sum-form PI-DRMDP-0.

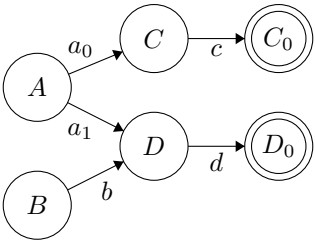

---

[5]Without loss of generality, the discussion is simplified by assuming $s_n$ will only appear at the end of a reward interval.

where for state $B, C, D$, there is only one action. The agent only needs to decide among $a_0, a_1$ at state $A$. $n$ is fixed as 2 and the initial state distribution $p_0(A) = p_0(B) = 0.5$. The reward has sum-form over the following per-step reward

$$r(C, c) = r(D, d) = 0$$
$$r(A, a_0) = 0.01, r(A, a_1) = 1$$
$$r(B, b) = -1$$

**Fact 8.** *For the above PI-DRMDP-0 and any initialization of the policy, value evaluation ($Q_\phi(s_t, a_t)$ in tabular form) via Eq. (1) with on-policy data and policy iteration w.r.t $Q_\phi(s_t, a_t)$ will converge to a sub-optimal policy with $\mathcal{J}(\pi) = -0.495\gamma$. In contrast, our algorithm can achieve the optimal policy with $\mathcal{J}(\pi) = 0$.*

*Proof.* For any policy $\pi$, we denote $p = \pi(a_1|A)$. Then, if we minimize Eq. (1) with on-policy samples, we will have $Q_\phi(C, c) = 0.01, Q_\phi(A, a_0) = 0.01\gamma$. Besides, as initial state distribution are uniform, we have

$$Q_\phi(D, d) = \frac{1}{2}(p - 1), \quad Q_\phi(A, a_1) = \frac{1}{2}\gamma(p - 1)$$

Since $Q_\phi(A, a_1) < Q_\phi(A, a_0)$ for any $p$, policy iteration will converges to a sub-optimal policy that $\pi(a_0|A) = 1$. The sub-optimal policy has $\mathcal{J}(\pi) = -0.495\gamma$. Noticing that this holds even if the initial policy is already the optimal.

For our algorithm, we have

$$Q_\phi(A, a_1) = \gamma Q_\phi(A, a_0, D, d) = \gamma$$

For any $\gamma > 0$ and any $p$, a single policy iteration will have the optimal policy with but the optimal has $\mathcal{J}(\pi^*) = 0$. $\qquad\square$

## B.7 DISCUSSION OF DRMDP DEFINITION

In this section, we compare DRMDPs with other related definitions about MDP delays. To be specific, we provide brief discussions on the differences between DRMDP (delayed reward MDP) with Delay MDP and Semi-MDP.

**Semi-MDP and Option Framework**: In Semi-MDP, actions are allowed to execute variable length of time. In DRMDP, actions are executed and re-planned at each time step while rewards are delayed for random length and are non-Marvian. Simply solving DRMDPs (or PI-DRMDPs) as Semi-MDPs will result in serious sub-optimal problems.

The middle ground between Semi-MDP and MDP is the Option Framework (Sutton, 1998), in which options are temporally extended action sequences and cen be learned, i.e., option policy, initial state set, termination state set. Contrarily, in DRMDP, the length of the signal interval (comparing to the length of the option execution time) is random and is determined by the environment. Furthermore, in option framework, rewards are still provided once very step (Markovian).

Option framework is often used on problems with hierarchical structures (Al-Emran, 2015), in which the whole policy is composed of a low-level option policy over actions and a high-level control policy over options. However, DRMDPs are formulated for general sequential decision making problems.

**Delay MDP**: Delay MDPs (Walsh et al., 2009; Derman et al., 2021) are used to characterize observation delay and action delay. For example, the observation inputs to the controller (e.g., a computer) is the observation of the remote agent (e.g., a drone) several milliseconds ago and the action executed at the remote agent is the output of the control policy several steps ahead. We have to emphasize that, in delay MDP, the rewards are still defined as per-step rewards and are coupled with the corresponding state-action pair, i.e., when the delay is a constant $m$, the transition $(s_t, a_t, r_t, s_{t+1})_a$ observed at the remote agent, is also the transition the controller received at step $t + m$, i.e., $(s_{t+m}, a_{t+m}, r_{t+m}, s_{t+m+1})_c$.

In DRMDP, there is no observation-delay or action-delay and there is no per-step rewards coupled with each state-action pair.

# C ADDITIONAL RESULTS

## C.1 DESIGN EVALUATION

Here, we show more experiments omitted in Section 4.1. To begin with, we compare different HC-decomposition algorithms ($\mathcal{Q}$-HC) with $\mathcal{Q}$-RNN and vanilla SAC on several sum-form PI-DRMDP tasks with different signal interval length distributions ($q_n$). The results are consistent with our discussions in Section 4.1, despite of the lengths and the distributions.

Table 2: Relative average performance in tasks with different signal interval length distribution $q_n$ over 4 environments: Hopper-v2, HalfCheetah-v2, Walker2d-v2, Ant-v2. $\delta(x)$ refers to a fixed interval length as $x$ and $U(a, b)$ refers a uniform distribution in range $[a, b]$. All results show the mean of 6-7 seeds and N.A. refers to no result due to limited computation resources.

| $q_n$ | SAC | $\mathcal{Q}$-RNN | $\mathcal{Q}$-HC-Singleton | $\mathcal{Q}$-HC-Pairwise-1 | $\mathcal{Q}$-HC-Pairwise-3 | $\mathcal{Q}$-HC-RNN |
|---|---|---|---|---|---|---|
| $\delta(10)$ | 0.62 | 0.61 | 0.91 | 0.89 | 0.92 | **0.98** |
| $U(10, 20)$ | 0.48 | 0.52 | 0.95 | 0.84 | 0.95 | **0.96** |
| $U(15, 20)$ | 0.48 | 0.61 | 0.89 | **0.99** | 0.95 | 0.93 |
| $\delta(20)$ | 0.51 | 0.53 | 0.91 | 0.93 | 0.98 | **0.99** |
| $\delta(40)$ | 0.30 | 0.38 | 0.90 | **0.94** | 0.85 | 0.87 |
| $\delta(60)$ | 0.17 | N.A. | 0.87 | 0.81 | **0.88** | N.A. |

Then, we test the HC-decomposition on *non* sum-form PI-DRMDP taks based on OpenAI Gym. Two *non* sum-form tasks are included: *Max* and *Square*. For *Max*,

$$r(\tau_{t:t+n}) = 10 \times \max_{i \in [t:t+n]} \{r(s_i, a_i)\}$$

Here, $r(s_i, a_i)$ is the standard per-step reward in each task. For *Square*, we denote $r_{\text{avg}}(\tau_{t:t+n}) = \frac{1}{n} \sum_{i \in [t:t+n]} r(s_i, a_i)$ and the reward is defined as follows.

$$r(\tau_{t:t+n}) = 4 \times \begin{cases} r_{\text{avg}}(\tau_{t:t+n}) & |r_{\text{avg}}(\tau_{t:t+n})| < 1. \\ \text{sign}(r_{\text{avg}}(\tau_{t:t+n})) \times r_{\text{avg}}^2(\tau_{t:t+n}) & |r_{\text{avg}}(\tau_{t:t+n})| \geq 1. \end{cases}$$

Noticing that these two tasks are still PI-DRMDP. The results are shown in Figure 5. Clearly, $\mathcal{Q}$-HC algorithms outperform $\mathcal{Q}$-RNN and vanilla SAC on all tasks.

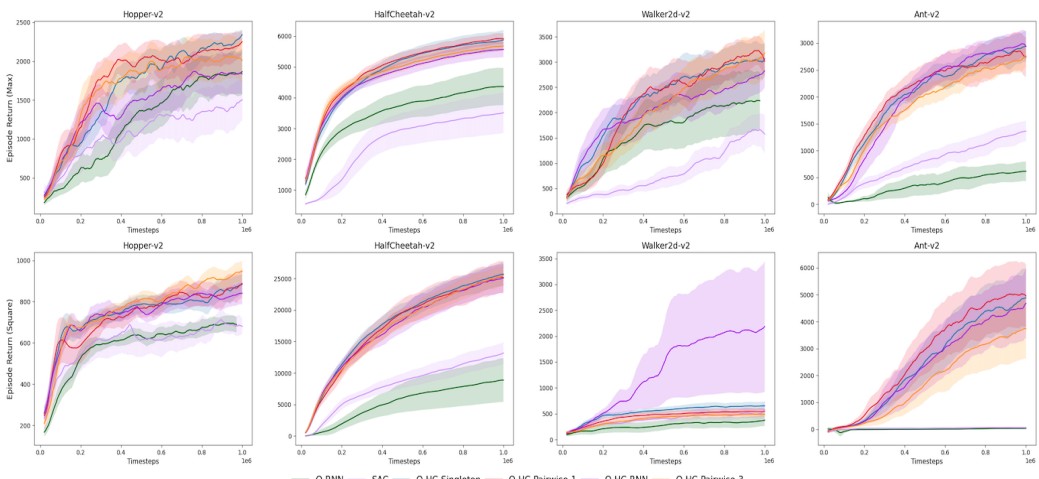

Figure 5: Experiments of *non* sum-form PI-DRMDP tasks. The first line shows the result of *Max* form and the second line shows the result of *Square* form. The learning curves are mean and one standard deviation of 6-7 seeds.

Moreover, we also implement TD3-based $\mathcal{Q}$-HC-Singleton, $\mathcal{Q}$-HC-Pairwise-1 and compare them with the vanilla TD3 and TD3-based $\mathcal{Q}$-RNN. The results still match our analysis of HC-decomposition in Section 3.2. However, we find that SAC-based variants consistently outperform

TD3-based variants. We think that the entropy maximization regularization is useful in the delayed reward environment, which prevents the policy getting trapped in local maximal points.

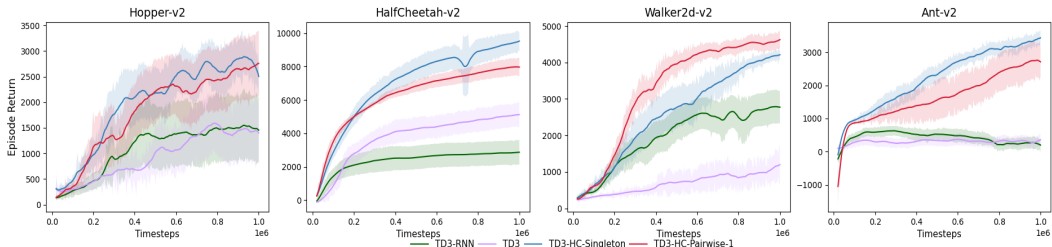

Figure 6: Learning curves of TD3-based $\mathcal{Q}$-HC-Singleton, $\mathcal{Q}$-HC-Pairwise-1, $\mathcal{Q}$-RNN and vanilla TD3. All curves show the mean and one standard deviation of 7 seeds. The result is consistent with that of SAC-based methods.

## C.2 ABLATION STUDY

In Figure 7, we show the ablation study on the regulation term $L_{\text{reg}}$ on all 4 tasks. We observe that in HalfCheetah-v2 and Ant-v2, the regulation plays an important role for the final performance while the regulation is less necessary for the others. Besides, for $\mathcal{Q}$-HC algorithms with complex H-component architectures (i.e., $\mathcal{Q}$-HC-RNN, $\mathcal{Q}$-HC-Pairwise-3), the regulation is necessary while for simple structure like $\mathcal{Q}$-HC-Singleton, regulation is less helpful. We suspect that the simple structure itself imposes implicit regulation during the learning.

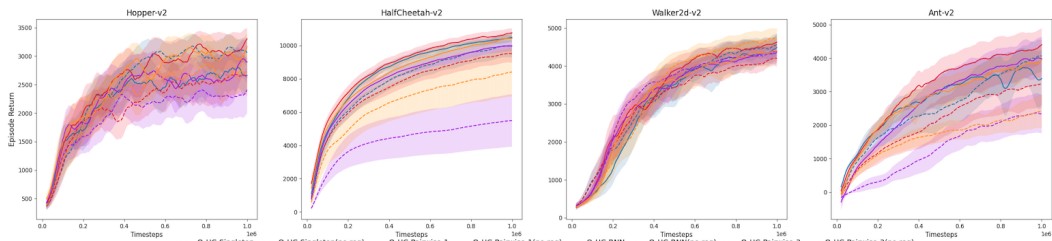

Figure 7: More ablations on $L_{\text{reg}}$ term. The task is a sum-form PI-DRMDP with $n$ uniformly drawn from 15 to 20. Dashed lines are the no regularization version of the algorithm $\mathcal{Q}$-HC with the same color. All curves show the mean and a standard deviation of 6-7 seeds.

Furthermore, we ablate the HC-decomposition in $\mathcal{Q}$-HC-Singleton and $\mathcal{Q}$-HC-Pairwise-1. Namely, we implement $\mathcal{Q}$-Singleton and $\mathcal{Q}$-Pairwise-1 with the similar architecture but *not* with HC-decomposition. For $\mathcal{Q}$-Singleton, we have

$$\mathcal{Q}(\tau_{t_i:t+1}) = \sum_{j \in [t_i:t+1]} b_\phi(\tau_{j:j+1})$$

and for $\mathcal{Q}$-Pairwise-1, we have

$$\mathcal{Q}(\tau_{t_i:t+1}) = \sum_{j \in [t_i:t]} c_\phi^1(\tau_{j:j+1} \circ \tau_{j+1:j+2}) + \sum_{j \in [t_i:t+1]} b_\phi(\tau_{j:j+1})$$

In Figure 8, we observe that these two ablations perform sluggishly and even worse than $\mathcal{Q}$-RNN (without any decomposition). This suggests the necessity of HC-decomposition for PI-DRMDP, i.e., the separated history part ($H$) and the current part ($C$), as the transition of the environment is still Markovian. Without the separation, a simple decomposition fails to assign the adequate credit to the last step action in $\tau_{t_i:t+1}$, thus results in errorneous policy update.

## C.3 COMPARATIVE EVALUATION

We show detailed results of our algorithms $\mathcal{Q}$-HC and previous baselines on sum-form PI-DRMDP tasks with general reward function. Please refer to Appendix B and Appendix A for

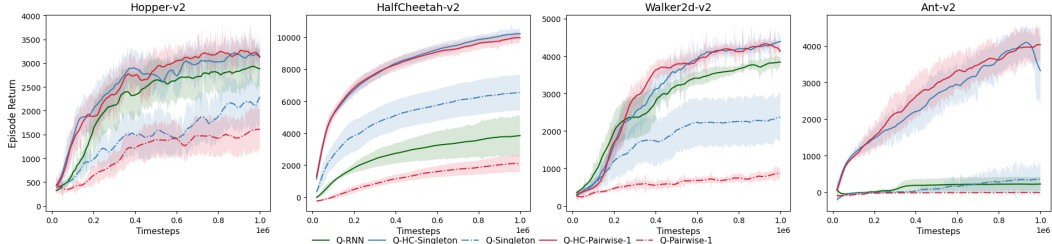

Figure 8: Ablation of HC-decomposition in $\mathcal{Q}$-HC-Singleton and $\mathcal{Q}$-HC-Pairwise-1. The task is a sum-form PI-DRMDP with $n$ fixed as 20. Dashed lines are two ablations $\mathcal{Q}$-Singleton and $\mathcal{Q}$-Pairwise-1. All curves show the mean and a standard deviation of 6 seeds.

detailed discussions. For environments with maximal $c$ overlapping steps, the reward function is formally defined as follows.

$$r_c(\tau_{t_i-c:t_i+n_i}) = \sum_{i \in [t_i-c:t_i+n_i-c]} r(s_i, a_i)$$

where $r(s_i, a_i)$ is the standard per-step reward. If $i < 0$, then $r(s_i, a_i) = 0$. In this definition, the reward is delayed by $c$ steps and thus the signal intervals are *overlapped*. In Figure 9, we show the relative average performance (over Reach-v2, Hopper-v2, HalfCheetah-v2, Walker2d-v2, and Ant-v2) of each algorithm w.r.t the Oracle SAC trained on the dense reward setting. Please refer to Appendix D for the exact definition of this metric. The results demonstrate the superiority of our algorithms over the baselines in the General Reward Function experiments. This supports our claim that our algorithmic framework and the approximation method (HC) can be extended naturally to the general definition, i.e., PI-DRMDP-c. Furthermore, we show the detailed learning curves of all algorithms in tasks with different overlapping steps $c$ in Figure 10.

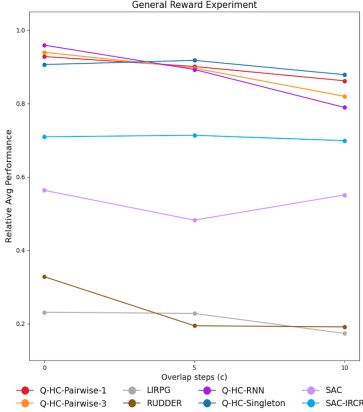

Figure 9: Relative Average Performance of our algorithms and the baselines on PI-DRMDP-c tasks with general reward functions. Each dot represents the relative performance of the algorithm over the PI-DRMDP-c task (average over Reach-v2, Hopper-v2, HalfCheetah-v2, Walker2d-v2, Ant-v2).

## D   EXPERIMENT DETAILS

### D.1   IMPLEMENTATION DETAILS OF OUR ALGORITHM

We re-implement SAC based on OpenAI baselines (Dhariwal et al., 2017). The corresponding hyperparameters are shown in Table 3. We use a smaller batch size due to limited computation power.

To convince that the superiority of our algorithms ($\mathcal{Q}$-HC-RNN, $\mathcal{Q}$-HC-Pairwise-K, $\mathcal{Q}$-HC-Singleton) results from the efficient approximation framework (Section 3.2) and the consistency

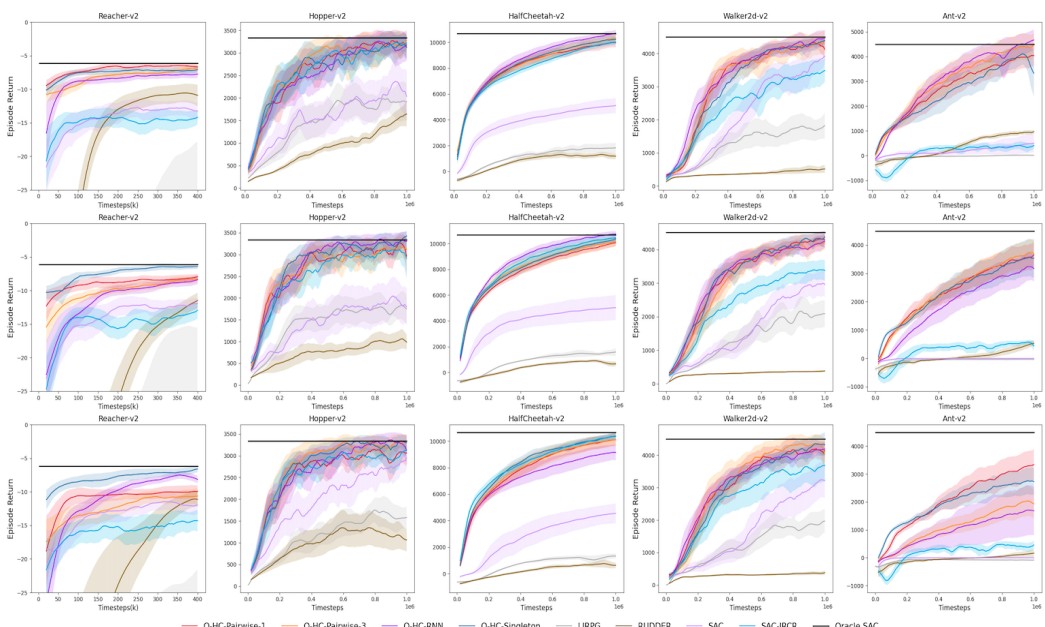

Figure 10: Learning curves of our algorithms $Q$-HC and several previous baselines. The first line refers to the task with $c = 0$. The second line refers to the task with $c = 5$. The third line refers to the task with $c = 10$. All curves show the mean and one standard deviation of 6-7 seeds.

Table 3: Shared Hyperparameters with SAC

| Hyperparameter | SAC |
|---|---|
| $\mathcal{Q}_\phi, \pi_\theta$ architecture | 2 hidden-layer MLPs with 256 units each |
| non-linearity | ReLU |
| batch size | 128 |
| discount factor $\gamma$ | 0.99 |
| optimizer | Adam (Kingma & Ba, 2014) |
| learning rate | $3 \times 10^{-4}$ |
| entropy target | $-|A|$ |
| target smoothing $\tau$ | 0.005 |
| replay buffer | large enough for 1M samples |
| target update interval | 1 |
| gradient steps | 1 |

with the theoretical analysis (Section 3.1), rather than from a better choice of hyper-parameters, we did not tune any hyper-parameter shared with SAC (i.e., in Table 3).

The architecture of different $H_\phi$s are shown as follows. For the regularization coefficient $\lambda$, we search $\lambda$ in $\{0.0, 0.05, 0.5, 5.0\}$ for each implementation of $\mathcal{Q}$-HC. $\lambda$ *is fixed for all tasks including the toy example.*

- $\mathcal{Q}$-HC-RNN. Each step's input $\tau_{t:t+1}$ is first passed through a fully connected network with 48 hidden units and then fed into the GRU with a 48-unit hidden state. $H_\phi(\tau_{t_i:t})$ is the output of the GRU at the corresponding step. We use $\lambda = 5.0$.

- $\mathcal{Q}$-HC-Pairwise-1. Both $c_\phi^0$ and $c_\phi^1$ are two-layer MLPs with 64 hidden units each. $\lambda = 0.5$.

- $\mathcal{Q}$-HC-Pairwise-3. All $c_\phi^i, i = 0, 1, 2, 3$ are two-layer MLP with 48 hidden units each. $\lambda = 5.0$.

- $\mathcal{Q}$-HC-Singleton. $b_\phi$ is a two-layer MLPs with 64 hidden units each. $\lambda = 0.05$.

As suggested by the theoretical analysis, we augment the normalized $t - t_i$ to the state input.

All high dimensional experiments are based on OpenAI Gym (Greg et al., 2016) with MuJoCo200. All experiments are trained on GeForce GTX 1080 Ti and Intel(R) Xeon(R) CPU E5-2630 v4 @ 2.20GHz. Each single run can be completed within 36 hours.

### D.2 IMPLEMENTATION DETAILS OF OTHER BASELINES

$\mathcal{Q}$-**RNN**. We use the same shared hyper-parameters as in Table 3. The only exception is the architecture of the GRU critic. Similar to $\mathcal{Q}$-HC-RNN, each state-action pair is first passed through a fully connected layer with 128 hidden units and ReLU activation. Then it is fed into the GRU network with 128-unit hidden state. The $\mathcal{Q}_\phi(\tau_{t_i:t+1})$ is the output of the GRU network at the corresponding step. We choose the above architecture to ensure that the number of parameters is roughly the same with our algorithm for fair comparison. The learning rate for the critic is still $3 \times 10^{-4}$ after fine-tuning.

**SAC-IRCR**. Iterative Relative Credit Refinement (Gangwani et al., 2020) is implemented on episodic reward tasks. In the delayed reward setting, we replace the smoothing value (*i.e.*, episode return) with the reward of the reward interval. We find this performs better than using the episode return.

### D.3 DETAILS FOR TOY EXAMPLE

**Point Reach.** As illustrated in Figure 4, the Point Reach PI-DRMDP task consists of a 100x100 grid, the initial position at the bottom left corner, a 10x10 target area adjacent to the middle of the right edge and a point agent. The observation for the point agent is its $(x, y)$ coordinate only. The action space is its moving speed $(v_x, v_y) \in [-1, 1]^2$ along the two directions. Since the agent can not observe the goal directly, it has to infer it from the reward signal. To relieve the agent from heavy exploration (not our focus), the delayed rewards provide some additional information to the agent as follows.

The grid is divided into 10 sub-areas of 10x100 along the x-axis. We denote the sub-areas (highlighted by one color each in Figure 4) as $S_i, i = [0 : 10]$ form left to right. In each sub-area $S_i$, it provides an extra reward and the reward $r^i$ indicates how far is the sub-area to the target (*i.e.*, $r^i = i$). Clearly, the bigger the reward is, the closer the sub-area is to the target area. Besides, reward interval length is fixed as 20 and the point agent is rewarded with the maximal $r^i$ it has visited in the interval together with a bonus after reaching the target and a punishment for not reaching. Namely,

$$r(\tau_{t:t+n}) = \max_{j \in [t:t+n]} \sum_{i=0}^{9} r^i \cdot \mathbf{1}(s_{j:j+1} \in S_i) + 10 \cdot (\mathbf{1}(reach\ target) - 1)$$

The task ends once the agent reach the target area and also terminates after 500 steps. Clearly. the optimal policy is to go straightly from the initial point to the target area without hesitation. The shortest path takes roughly 95 steps. The learning curves in Figure 4 show the mean and half of a standard deviation of 10 seeds. All curves are smoothed equally for clarity.

**Heatmap.** In Figure 4 right, the heatmap visualizes the value of $b_\phi(s_t, a_t)$ on the whole grid. We select the $b_\phi$s after the training has converged to the optimal policy. To be specific of the visualization, for each of the 10x10 cells on the grid, $s_t$ in $b_\phi(s_t, a_t)$ is selected as the center of the cell and $a_t$ is sampled from $\pi(\cdot|s_t, 0)$. Additionally, in Figure 11, we visualize $b_\phi(s_t, a_t)$ similarly of $\mathcal{Q}$-HC-Singleton without $L_{reg}$. Clearly, Figure 11 shows the similar pattern as Figure 4 right.

### D.4 DETAILS FOR VARIANCE ESTIMATION

The experiment in Figure 1 is conducted in the following manner. First, we train a policy $\pi_\theta$ with $\mathcal{Q}$-HC-Singleton and collect all the samples along the training to the replay buffer $D$. Second, two additional critics are trained concurrently with samples uniformly sampled from $D$. One critic uses the GRU architecture (i.e., $\mathcal{Q}$-RNN) and the other uses $\mathcal{Q}$-HC-Singleton structure. Most importantly, since these two critics are not used for policy updates, there is no overestimation bias (Fujimoto et al., 2018). Thus, instead of using the CDQ method Fujimoto et al. (2018), these two critics are trained via the method in DDPG (Silver et al., 2014). With a mini-batch from $D$, we compute the gradients on the policy's parameters $\theta$ in Eq. (5) and Eq. (7) from the two extra critics respectively. Noticing that these gradients are not used in the policy training.

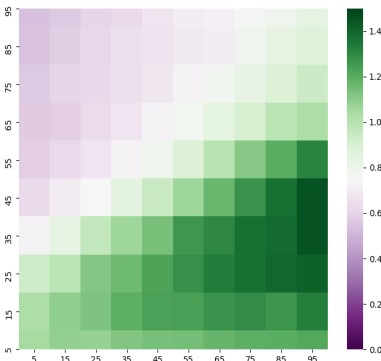

Figure 11: Visualization of $b_\phi(s_t, a_t)$ of $\mathcal{Q}$-HC-Singleton without regulation term.

Then, we compute the sample variance of the gradient on each parameter $\theta_i$ in the policy network. The statistics (y-axis in Figure 1) is the sum of variance over all parameters in the final layer (*i.e.*, the layer that outputs the actions). The statistics is further averaged over the whole training and scaled uniformly for clarity.

### D.5 DETAILS FOR RELATIVE AVERAGE PERFORMANCE

Here, we formally introduce the Relative Average Performance (*RAP*) metric in Table 2, Figure 9, and Figure 3. The relative average performance (*RAP*) of algorithm $A$ on task set $E$ is defined as the average of *RAP* over tasks in $E$. *RAP* of each task is the episodic return after 1M steps training normalized by the episodic return of Oracle SAC at 1M steps (trained on dense reward environment). The only exception is Reach-v2, in which the episodic returns are added by $50$ to before the normalization.

