# OpenReview forum: "Off-Policy Reinforcement Learning with Delayed Rewards"
_ICLR.cc/2022/Conference — ICLR 2022 Submitted_

### Official Review · Reviewer_fukM · 2021-10-27

**Correctness:** 2
**Technical Novelty And Significance:** 2
**Empirical Novelty And Significance:** 2
**Recommendation:** 3
**Confidence:** 4

**Main Review:**

The paper opens with statements on how delay in MDPs is an unexplored territory. However, the authors seem to ignore both mature and recent works that address it thoroughly and reach conclusions that may affect the findings in this paper. Two examples are [1] and [2]. I find [2] even more relevant, since like in this reviewed paper, it also proposes a new Q-function definition and discusses the related non-stationarity.

Next, even though the grammar is fine, I find the paper hard to follow. Specifically, the past-invariance introduced in Def. 2 that constitutes the basis for the consecutive results is not clear to me. Does it really assume the reward is independent of past decisions but only the current one? How is that realistic in a delayed reward setup, where the history is exactly what yields the sequence of present reward? Under such a strong assumption, I believe the result in Fact 2 is almost trivial.

In similar fashion, the H-C decomposition in Sec. 3.2 seems extremely artificial. If I understand correctly, it assumes the Q-function decomposes to two additive parts of past and present. From an engineering POV, I can see how it is useful. But how does this make sense in a delayed reward setup where everything depends on past decisions? I would expect more formal and informal (intuitive) justifications for such a strong assumption. Note also that making this assumption on the Q-function is stronger than making it on the reward.

Additional comments/questions:
1. First sentence in p.3 -- why is it bizarre that the proposed PI-DRMDP requires the optimal policy to consider history? Can the authors provide an example of such a case? If not, can they show the opposite and provide a proof that there is always an optimal history that is history independent? This is expected especially as the authors are those who propose this model and later rely on this assumption. Also, what information exactly is required for the policy to be optimal? I general, the usage of the word "bizarre" without any justification is bothering.
2. I find the explanations below Sec 2.2 to be confusing, especially the paragraph below the display equation and the multiple behavior policies that "come out of nowhere".


[1] Walsh, Thomas J., et al. "Learning and planning in environments with delayed feedback." Autonomous Agents and Multi-Agent Systems 18.1 (2009): 83-105.

[2] Derman, Esther, Gal Dalal, and Shie Mannor. "Acting in Delayed Environments with Non-Stationary Markov Policies." International Conference on Learning Representations (2021).

**Summary Of The Paper:**

The paper studies the problem of delayed reward, which imposes non-Markov behavior that hinders performance of "standard" RL algorithms. It then proposes a new Q-function definition based on segments of reward, together with a decomposition to historical-current parts. Lastly, the authors add experimental comparisons to several recently published methods.

**Summary Of The Review:**

The paper makes strong assumptions on MDPs with delayed reward that do not align well with knowledge from previous works. Instead of trying to compare and articulate the exact differences and conclusions of their work and previous literature, the authors briefly list some existing works at the end of the paper in an independent section without such warranted explanations. While I appreciate the vast experimental part of the paper with comparison to several other methods, the rest of the paper is not well justifying many of its statements and does not convince that the overall approach makes sense.

---

### Official Review · Reviewer_oufu · 2021-11-02

**Correctness:** 2
**Technical Novelty And Significance:** 3
**Empirical Novelty And Significance:** Not applicable
**Recommendation:** 3
**Confidence:** 4

**Main Review:**

Strengths\
         The idea is pretty new and interesting.
         It seems from the empirical results that the proposed algorithm performed well in the tested domains.

The main weakness of the paper is that it is very difficult to understand. There are just so many parts that are confusing to me.\
         a) Generally speaking, it seems to me that this paper tries to present so many different things in one paper without making each of them clear to the readers. For example, the main theoretical contribution of the paper, the newly proposed decision process, once defined, is only briefly discussed in section 3.1. It is hard to tell the commonalities and differences between the new decision process and classic MDP. And I would also suggest the authors compare the proposed decision process with Semi-MDP because they seem to be closely related. The other example I can give is that the paper derives a policy improvement theorem and then directly jumps to a deterministic policy gradient (equation 5), which uses continuous actions while all the definitions and results derived up to that point are for discrete actions. Then the paper starts to talk about neural network implementation of the policy gradient. It is like algorithms, theories, and implementations are all mixed together without a clear explanation of each of them. \
         b) Each policy in \Pi_s is a function of s_t, a_t, and t - t_i. But the action-value function takes the entire trajectory segment as input, which entails more information than s_t, a_t, and t - t_i. Do we really need to let the action-value function take the entire trajectory segment as input in order to get the best policy in \Pi_s?\
         c) Given that the authors want to find the optimal policy within \Pi_s, why not just combine s_t and t - t_i and treat them as a state? You would convert the problem to the normal MDP problem in this way.\
         d) Some sentences are confusing: e.g., page 4 "Consequently, Q can be found precisely via the minimization for tabular expressions". Why is this important? This is true when you know \hat Q, but from the paper, it seems that you don't know it right?\
         e) Frankly speaking, the notations are not easy to follow. You might want to consider other ways to make it easier for readers to understand.



**Summary Of The Paper:**

This paper proposed a decision process that is Markov in state transitions but non-Markov in reward. The paper argued that this new decision process is a suitable model for applications where rewards are delayed. For this decision process, the paper proposed analogues of action-value function and policy improvement theorem. It further proposed to decompose the action-value into two parts, one of which only depends on the current state-action pair while the other one depends on the history. It then proposed an algorithm that extends SAC and estimates the two parts separately. Empirical results showed improved performance over SAC on delayed reward versions of several continuous control problems.

**Summary Of The Review:**

I suggest rejecting the paper because of the weaknesses listed in my main review. I like the idea of the paper and would encourage the authors to keep working on it. But the paper in its current form is not ready to be published from my point of view.

---

### Official Review · Reviewer_mJkz · 2021-11-03

**Correctness:** 3
**Technical Novelty And Significance:** 2
**Empirical Novelty And Significance:** 2
**Recommendation:** 3
**Confidence:** 4

**Main Review:**


- What is the relation between $t_i$ and $t+1$? The definition of $t_i$ only says that it is the first step of the interval $i$, but does not have any dependency on $t$ itself. Throughout the paper, the range of $t_i:t+1$ is used but I do not know if I understand this range properly.

- It felt that under dynamics being Markovian and only rewards being Markovian, then additionally enforcing past invariance condition will result in even the standard Q-learning to not having the fixed point bias. But authors provide an interesting example later on confirming the fixed point bias, if the non-Markovian structure is not accounted for. Perhaps this result can be alluded to earlier on in the paper.

- It is also worth mentioning that policy gradient, with full Monte-Carlo roll-outs can deal with any form of non-Markovian structure to provide exact policy evaluation and policy improvement (it can be seen from slide 9 here: http://rail.eecs.berkeley.edu/deeprlcourse-fa17/f17docs/lecture_4_policy_gradient.pdf). Further, using importance sampling it can be made to work in the off-policy setting. Of course, it may not find the optimal policy if it is not within the policy class. The problem occurs when using Q-learning (or actor-critic methods) that rely on the Markov structure through the Bellman equation to estimate the Q-values.

- However, I am not sure what the novelty of the proposed method is, both theoretically and practically. How is the proposed method different from just naively concatenating the history of the last n steps in the state? (This might also clarify my confusion about $t_i$ and $t+1$), given that the reward only depends on the signal interval $n$. One can now consider a new MDP that has state = concatenation of the past $n$ steps, and therefore Markovian in this new state space. Using this state-space, all the existing policy improvement results will hold directly. Of course, one problem here is about knowing what is the value of $n$, but I am not sure how is the proposed method handling that either? Is it assumed that the value of $n$ for any interval is provided to the agent?  (Yes, for the proposed method, the policy does not concatenate recent history, but Q still does)

- Another question worth addressing is which step fails in the final algorithm if even the states are Markovian (but past invariance still holds) ?

- Policy gradient equations are wrong everywhere in the paper (Eq 2, 5, and 7). It needs (a) distribution correction terms to account for the off-policy sampling under $D$ instead of $\pi$, and (b) is also missing the $\gamma$ terms in the gradient (Thomas, Philip. "Bias in natural actor-critic algorithms." International conference on machine learning. PMLR, 2014).

- For experiments, the paper mentions "To convince that the superiority of our algorithms..... we did not tune any hyper-parameter shared with SAC". This statement is very weak and makes the empirical results less meaningful. The empirical results only show that _for this given choice of hyper-parameters_ the proposed method is better. Further, there are additional parameters for the proposed method which are indeed tuned :(

- It is worth calling out before each Fact/Proposition that a tabular setting is being considered for that given result. Further, the proofs in the appendix have lots of writing errors and should be presented better.


**Summary Of The Paper:**

Authors look at a problem where the rewards are non-Markovian (but dynamics are Markovian). They show how naively using existing methods on this setup may result in sub-optimal policies and then provide a method based on considering recent history as the augmented state.



**Summary Of The Review:**

Paper falls a little short on both theoretical/methodological novelty and rigorous experimentation.

---

### Official Review · Reviewer_6XaH · 2021-11-03

**Correctness:** 3
**Technical Novelty And Significance:** 2
**Empirical Novelty And Significance:** 2
**Recommendation:** 5
**Confidence:** 4

**Main Review:**

1. This paper proposes a new setting of delayed reward, namely DRMDP. It seems unclear what real problem can be exactly modeled by this setting. Moreover, Past-Invariant Delayed Reward Markov Decision Process seems to impose an additional assumption that the past does not affect the future. What application can be modeled by this model? What does path-invariance mean for the distribution of signal interval length? What $q_n (\cdot)$ meet the assumption?

2. This paper proposes to focus on a restricted policy class $\Pi_s$. How good is it? Even if we achieve the globally optimal policy, it does not mean we find a "good" policy because the policy can be very small such that all policies in this class are bad.

3. For $Q^{\pi}$ to be the solution to the least-squares loss in (1), what's the assumption on the behavior policy? Does it also belong to $\Pi_s$? How to choose $\hat Q$? Is it possible to show that $Q^\pi$ is the fixed point of a Bellman operator for the policy evaluation problem?

4. How to conduct policy improvement step for $\Pi_s$?

5. What is the reasoning behind HC-decomposition? Why use an additive model instead of using a big GRU that takes everything as the input?

6. This work is related to bandit and RL with stochastic delays:

- Learning in Generalized Linear Contextual Bandits with Stochastic Delays
- Gradient-free Online Learning in Continuous Games with Delayed Rewards
- Distributed Asynchronous Optimization with Unbounded Delays: How Slow Can You Go?

**Summary Of The Paper:**

This work considers a modified MDP model where the rewards are subject to random delays. Specifically, the timesteps are divided into random batches and the agent observes the cumulative rewards of the batch when it ends. For such a model, Markov property does not hold and classical methods do not apply. The authors propose a policy iteration method for learning the optimal policy within a restricted policy class. In particular, the authors consider a more restricted model, named Past-Invariant Delayed Reward Markov Decision Process, and a policy class with an augmented parameter that indicates the current time step in the signal batch. In such a setting, the authors show that, under the tabular case, the true value function minimizes the mean-squared error of the policy evaluation problem, and the policy iteration step yields an improved policy. Furthermore, to enhance computational efficiency, the HC-decomposition is utilized in the Q-function.

**Summary Of The Review:**

This work proposes a new setting of MDP models with stochastic delays. But the assumption of the MDP model is not well-justified -- it is unclear what signal interval distribution justifies the assumption. Also, the policy class might be restricted and it is unclear how to conduct policy improvement within the policy class.

---

### Decision · Program_Chairs · 2022-01-20

**Decision:**

Reject

**Comment:**

The paper introduces an interesting new model for MDPs, where the time is divided into random segments, and at the end of each segment the cumulative reward for the given segment is communicated to the agent. Some theoretical results with a policy improvement algorithm, as well as a more practical algorithm are presented. While the reviewers valued these contributions, they all had issues with the presentation of the paper.

These presentation issues make the paper extremely hard to follow -- this was a problem for all reviewers, and I also verified it myself. The reviewers also raised issues regarding the experiments, where the algorithms should be tuned properly to be able to draw valid conclusions.

While unfortunately the above issues prevent me from recommending acceptance of the paper, the authors are strongly encouraged to revise their paper and resubmit to the next venue, with a special emphasis on making the presentation proper. There are several problems/recommendations mentioned in the reviews which will certainly help in this regard (I would also add that special care should be made that everything is defined properly, e.g., the equation for your policy iteration should appear in the main text not in a proof in the appendix, or $\hat{Q}_\phi$ should be defined, etc.).